# DeepSeek-Prover-V1.5: Harnessing Proof Assistant Feedback for Reinforcement Learning and Monte-Carlo Tree Search

**Huajian Xin**[†][*]  **Z.Z. Ren**[*]  **Junxiao Song**[*]  **Zhihong Shao**[*]  **Wanjia Zhao**  **Haocheng Wang**
**Bo Liu**  **Liyue Zhang**  **Xuan Lu**  **Qiushi Du**  **Wenjun Gao**  **Haowei Zhang**
**Qihao Zhu**  **Dejian Yang**  **Zhibin Gou**  **Z.F. Wu**  **Fuli Luo**  **Chong Ruan**
DeepSeek-AI     [†]The University of Edinburgh

## Abstract

Lean is an advanced proof assistant designed to facilitate formal theorem proving by providing a variety of interactive feedback. In this paper, we explore methodologies to leverage proof assistant feedback to augment the capabilities of large language models in constructing formal proofs. First, we deploy online reinforcement learning using Lean verification outcomes as the reward signal to improve the proof completion policy. This straightforward approach shows great promise in enhancing the model's alignment with the formal verification system. In addition, we propose RMaxTS, a variant of Monte-Carlo tree search that employs an intrinsic-reward-driven exploration strategy to generate diverse proof paths. The tree structure is organized to represent the transitions of intermediate tactic states, extracted from the compilation messages given by Lean's tactic mode. The intrinsic reward is constructed to incentivize the discovery of novel tactic states, which helps to to mitigate the sparse-reward problem inherent in proof search. These techniques lead to a more efficient planning scheme for formal proof generation, achieving new state-of-the-art results on both miniF2F and ProofNet benchmarks.

## 1 Introduction

Recent advancements in large language models have significantly influenced mathematical reasoning and theorem proving in artificial intelligence. Despite notable progress in natural language domains, language models still encounter substantial challenges in formal theorem proving, *e.g.* using Lean (Moura & Ullrich, 2021) and Isabelle (Paulson, 1994), which requires rigorous derivations satisfying formal specifications of the verification system. Even advanced models like GPT-4 (OpenAI, 2023) struggle with complex formal proofs, underscoring the intricate nature of both the coding and the mathematics involved. Language models in formal theorem proving typically employ two strategies: proof-step generation (Jiang et al., 2022a; Lample et al., 2022; Yang et al., 2023; Wu et al., 2024) and whole-proof generation (Zhao et al., 2023; Wang et al., 2023a). The proof-step generation approach is motivated by the interactive nature of Lean's tactic mode, in which the compiler provides the access to the tactic state, *i.e.*, a structured representation summarizing the current status of the proof, including all the relevant information such as the local context of hypotheses and pending goals. Given the intermediate tactic state, the proof-step generation approach predicts each subsequent tactic and verifies it using the formal verifier to obtain updated information about the current tactic state. This interactive process often employs tree search techniques to compose valid proofs through several iterations of tactic generation (Polu & Sutskever, 2020). In contrast, the whole-proof generation approach treats the construction of formal proofs as a general code completion task. This branch of methods aims to generate the entire proof code based on the theorem statement and perform verification only at the end of the generation process. The simplicity of the whole-proof generation paradigm has been proven to offer high scalability (Xin et al., 2024) from the perspectives of both model training and inference deployment. In addition, the whole-proof generation model is trained to perform long-term planning for theorem proving, facilitating the integration and utilization of the model's capabilities in natural language mathematical reasoning (Jiang et al., 2022b).

---

[*]Core contributors

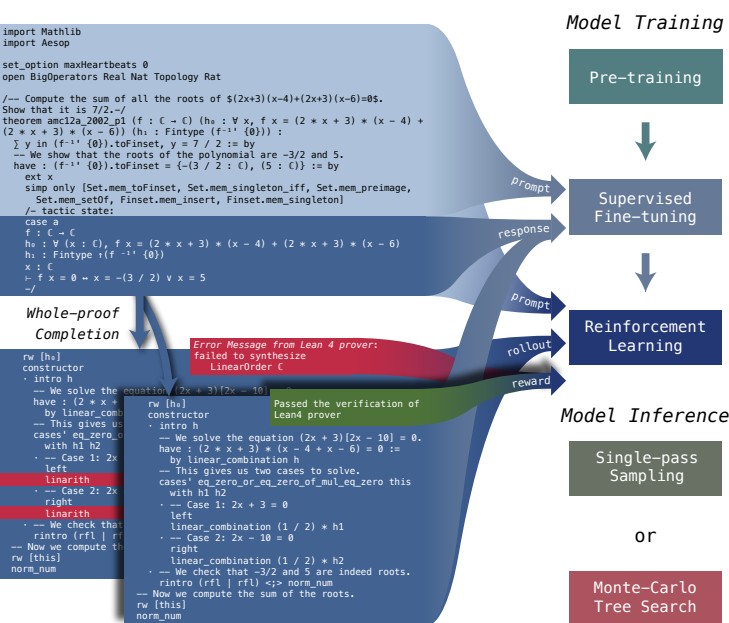

Figure 1: **Overall Framework of DeepSeek-Prover-V1.5.** During supervised fine-tuning, the model receives an incomplete theorem proof ending with a tactic state comment keyword. The model is trained to predict the content of this tactic state (auxiliary objective) and complete the subsequent proof steps (main objective). In the reinforcement learning stage, given an incomplete theorem proof and ground-truth tactic state from the Lean prover, we roll out the fine-tuned model to generate multiple proof candidates, which are then verified by the Lean prover. The verification results for these candidates are used as binary rewards to further optimize the model and enhance its alignment with the formal specifications of the verification system. For model inference, we decompose the generated proof into a series of tree nodes, appending intermediate tactic states extracted from the Lean prover, thereby establishing an interactive proof search paradigm.

In this paper, we present a unified approach that combines the strengths of both proof-step and whole-proof generation paradigms. We begin by training a whole-proof generation model, incorporating several auxiliary tasks to enhance its capabilities in mathematical reasoning and long-horizon planning, meanwhile empowering it to recognize information from Lean's proof assistant feedback. The model is named DeepSeek-Prover-V1.5, as it builds upon the prior work of DeepSeek-Prover-V1(Xin et al., 2024). We then employ a truncate-and-resume mechanism to decompose the whole-proof generation into a tactic-level proof search scheme. Figure 1 presents an illustration of our approach. The process begins with standard whole-proof generation, where the language model completes the proof code following the theorem statement prefix. The Lean assistant then verifies this code. If an error is detected, the code is truncated at the first error message, and any subsequent code is discarded. The successfully generated proof code is then used as a prompt for the generation of next proof segment. The latest tactic state from the Lean prover is appended at the end of the prompt as a comment block to provide intermediate guidance for the construction of long proofs. Notably, our method is not restricted to resuming from the last successfully applied tactic. We formalize the truncate-and-resume mechanism within the framework of Monte-Carlo tree search (MCTS; Coulom, 2006) in which the truncation points are scheduled by the tree search policy. In addition, we propose a novel reward-free exploration algorithm for MCTS to address the reward sparsity issue of proof search. We assign the tree search agent intrinsic motivation, *a.k.a.* curiosity (Schmidhuber, 2010), to extensively explore the tactic state space. These algorithmic modules extend the functionality of our whole-proof generation model to become a flexible tool for interactive theorem proving, which can effectively utilize the proof assistant feedback and generate diverse solution candidates. In experiments, we demonstrate substantial improvement of our proposed approach over baseline models, achieving new state-of-the-art results on the test set of the high school level miniF2F benchmark (63.5%) and the undergraduate level ProofNet benchmark (25.3%).

## 2 RELATED WORK

**Reinforcement Learning for Theorem Proving.**    Numerous prior research efforts have explored modeling the interaction interface with the proof assistants as a Markov Decision Process (MDP), leveraging various reinforcement learning techniques, such as using policy gradient (Zombori et al., 2021; Crouse et al., 2021) and deep Q-learning (Fawzi et al., 2019), and involving a wide range of formal verification systems, including E (McKeown & Sutcliffe, 2023), Coq (Kusumoto et al., 2018), and HOL4 (Gauthier, 2020). A common choice for the reward signal is a binary indicator denoting whether the proof has been completed. Fawzi et al. (2019) designed a temporal-difference reward assignment according to problem structure of solving polynomial inequality. Aygün et al. (2022) generalized the idea of hindsight experience replay (Andrychowicz et al., 2017) from goal-reaching control to formal theorem proving, which enriches the reward supervision.

**Tree Search for Theorem Proving.**    Integrating supervised models with search algorithms is a classical paradigm for automated theorem proving (Rawson & Reger, 2019; 2021; Zhang et al., 2024). For proof-step generation models, the most widely applied search strategy is best-first search (Yang et al., 2023), in which search branches are prioritized based on the cumulative log-likelihoods of the generated tactics. Lample et al. (2022) developed a specialized Monte-Carlo tree search algorithm tailored for the Lean theorem prover, in which subgoal branches are represented as hyperedges. The model training and tree search procedures are integrated similarly to the algorithmic framework of AlphaZero (Silver et al., 2018). Beyond the tactic-level tree abstraction, Wang et al. (2023b) investigated the effectiveness of using a proof-level value function in proof tree search, demonstrating that incorporating the entire proof as context is more effective than using a tactic-level state representation.

## 3 LEARNING TO UTILIZE PROOF ASSISTANT FEEDBACK

### 3.1 SUPERVISED FINE-TUNING

In this section, we explore the methodology and processes involved in the supervised fine-tuning (SFT) of DeepSeek-Prover-V1.5. Specifically, we incorporate intermediate tactic state information as an auxiliary prediction task to support the truncate-and-resume mechanism used in Monte-Carlo tree search. In addition, we augment the proof dataset from DeepSeek-Prover-V1 (Xin et al., 2024) by adding detailed explanatory comments. This enhancement aims to improve the alignment between natural language descriptions and Lean 4 code, thereby facilitating better formal mathematical reasoning. We refer to the resulting model as DeepSeek-Prover-V1.5-SFT, which is a 7B dense model trained on around 20B tokens. Details of data processing are described in Appendix A.2.

**Prompt Augmentation with Tactic State Information.**    To implement the truncate-and-resume mechanism for Monte-Carlo Tree Search, we needed to extract tactic information from the code generated by the model. We enhanced the Lean REPL (Read-Eval-Print Loop; Leanprover Community, 2023) with data extraction tools from the LeanDojo (Yang et al., 2023) project. This allowed us to extract tactic information in triples, which include the position of each tactic, as well as the tactic states before and after its application. This information helps us identify the specific tactic code that triggers verification errors (used in the expansion step for tree search, see Section 4.2). For each tactic in a generated valid formal proof, we insert the tactic state returned by the verifier as a comment "/- tactic state: ... -/". During training, we include all tokens following the leading prompt "/- tactic state: " as responses to calculate the supervised fine-tuning loss, while the tokens before this comment is used as prompts and do not contribute to the training loss calculation, *i.e.*, we construct an auxiliary task for the prover model to predict the current tactic state.

**Thought-augmented Proof Generation.**    Similar to Lean-STaR (Lin et al., 2024), which performs isolated chain-of-thought reasoning (Wei et al., 2022; Feng et al., 2023) before generating each proof step, we integrate this reasoning procedure directly as comments within the proof code. We use the DeepSeek-Coder V2 236B (Zhu et al., 2024) to enhance existing data in DeepSeek-Prover-V1 in two ways: first, by inserting a complete natural language solution at the beginning of the proof block, and second, by alternately inserting specific natural language steps for corresponding Lean tactics. Training the model with this data format enforces it to propose complete mathematical reasoning at the beginning of the proof block and detailed step planning before each

tactic. This approach successfully develops new behaviors, employing delicate mathematical thinking to guide the generation of tactics. In the training data, two distinct guiding prompts are used to differentiate between the CoT (Chain of Thought) mode and the non-CoT mode for proof code completion. Examples of input and output in both modes can be found in Appendix G.

**Discussion.** The primary purpose of implementing these data processing procedures during the SFT phase is to optimize the model's performance for downstream inference-time strategy. When applying tree search for proof generation, we leverage the model's ability to utilize proof assistant feedback while performing chain-of-thought reasoning. For an incomplete proof, we first append a comment block containing the ground-truth tactic state extracted from the Lean assistant, and then the model would perform chain-of-thought based on the full context information (see Figure 2). This procedure emulates the strategy employed by human experts when interacting with a formal proof assistant, combining both real-time feedback from the assistant and logical reasoning to iteratively refine and extend the proof. The improvements on inference-time performance are continuously fed back into SFT through *expert iteration* (Polu & Sutskever, 2020). The final SFT dataset is curated from challenging problems solved through tree search, aiming to fundamentally enhance the model's capabilities.

## 3.2 REINFORCEMENT LEARNING FROM PROOF ASSISTANT FEEDBACK

Reinforcement learning (RL) has been proven effective in enhancing the mathematical reasoning capabilities of supervised fine-tuned language models (Shao et al., 2024). To further advance DeepSeek-Prover-V1.5-SFT, we incorporate a reinforcement learning phase, resulting in the model DeepSeek-Prover-V1.5-RL. This phase leverages RL to enhance performance based on verification feedback from the Lean 4 prover. The specifics of this RL process are detailed below. Detailed training setting and hyper-parameters refer to Appendix A.3.

**Reinforcement Learning Algorithm.** We employ Group Relative Policy Optimization (GRPO; Shao et al., 2024) as our RL algorithm, which has demonstrated superior effectiveness and efficiency compared to PPO (Schulman et al., 2017), primarily because it eliminates the necessity of training an additional critic model. Specifically, GRPO samples a group of candidate proofs for each theorem prompt and optimizes the model based on the relative rewards of the outputs within the group. To ensure that both correct and incorrect proofs are included in the rollout candidates, we select a subset of theorem statements with appropriate difficulty from the supervised fine-tuning dataset as training prompts. These selected prompts are chosen based on the rule that DeepSeek-Prover-V1.5-SFT achieves a moderate success rate in generating correct proofs over multiple attempts. After filtering, we retain approximately 4.5k unique theorem statements. Each theorem is prefixed with both CoT and non-CoT guiding prompts to enhance the model's proof generation capabilities in both modes. The reward function is naturally given by the formal verification system, *i.e.*, each generated proof receives a reward of 1 if verified as correct, and 0 otherwise. The complete RL phase processes approximately 1.5B tokens.

**Discussion.** In Lean's tactic mode, proofs are constructed through a sequence of tactics that transform the proof state. This sequential nature introduces the risk of compounding errors (Ross et al., 2011), where a single misinterpretation can lead to significant deviations from a valid proof path. More specifically, the whole-proof generation model may have incorrect believes on intermediate tactic states when generating long proofs. Online reinforcement learning has been proven to be an effective method for mitigating compounding errors in the extrapolation setting of model inference (Fujimoto et al., 2019). By continuously interacting with the environment and receiving feedback in real-time, the model is able to refine its decision-making policy and reduce the bias inducted from the supervised dataset.

## 4 EXPLORATION-ORIENTED MONTE-CARLO TREE SEARCH

### 4.1 TACTIC-LEVEL TREE ABSTRACTION

To implement the tree search method in the whole-proof generation setting, we introduce a proof tree abstraction to define the tailored state and action space, leveraging a truncate-and-resume mechanism. Roughly following the paradigm of Yao et al. (2023), we begin by decomposing an incomplete proof into a sequence of tree nodes that correspond to individual proof steps, and then we utilize the

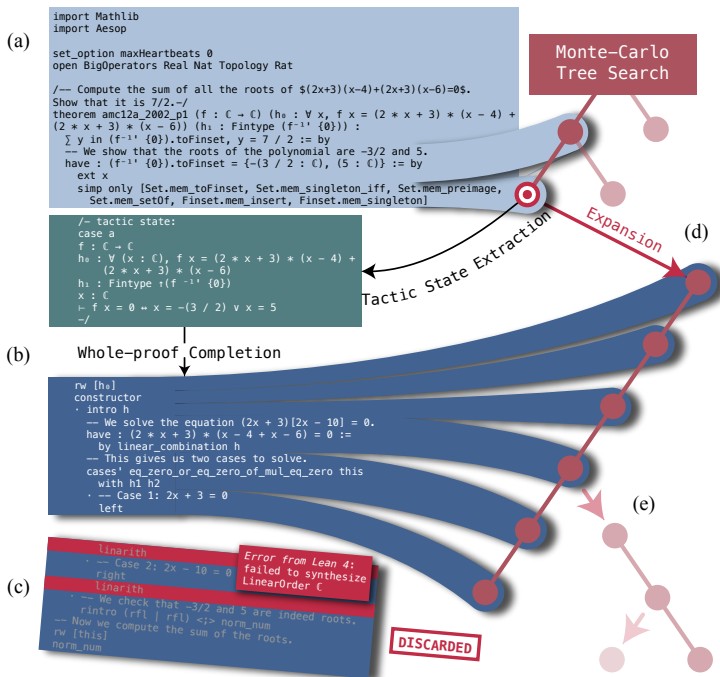

Figure 2: **Truncate-and-Resume Mechanism in the Expansion Step of MCTS.** (a) After selecting a node, we trace its corresponding incomplete proof code prefix, which includes the file header, initial statement, and successfully applied tactics from the ancestor nodes. (b) The language model then generates the subsequent proof based on this prefix along with a comment block containing the current tactic state. (c) The combined proof code (prefix and newly generated code) is verified by the Lean 4 prover. If no errors are found, the tree-search procedure terminates. If errors are detected, we truncate the newly generated code at the first error message, discard the subsequent code, and parse the successful portion into tactics. (d) Each tactic is added as a new node in the search tree, extending a chain of descendants beneath the selected node. (e) Once the tree updates are complete, the next iteration of expansion begins by selecting an alternative candidate node, which is not limited to leaf nodes. This process repeats until a correct proof is found or the sample budget is exhausted.

partial content stored in these tree nodes to continue the proof generation process. Figure 2 illustrates the process of constructing a proof search tree from whole-proof generation.

**Truncate: Proof Decomposition into Tree Nodes.**    We construct the proof search tree at the tactic level, where each tree edge represents a single transition step of the tactic state. Initially, we submit the entire proof the model generated to the Lean prover to parse it into tactics. We then truncate the proof at the earliest verification error, ensuring that all subsequent tactic codes can be successfully applied to advance the proof towards the desired theorem. The tactic codes are segmented into several code fractions, each containing a valid tactic code and its associated chain-of-thought comments, corresponding to a single tree edge that represents a tactic state transition. Through this abstraction, each tactic code is converted into a series of tree nodes, forming a path from the root to a specific node.

**Resume: Proof Generation from a Tree Node.**    In Lean 4, different tactics can lead to the same tactic state, meaning each node in our proof tree can correspond to various tactic codes that achieve the same outcome. To handle this, we store a set of these equivalent tactic codes at each node. When the tree search agent expands a node, it randomly selects one tactic to use as a prompt for the language model. This prompt includes the incomplete proof code ending with the chosen tactic and the tactic state information from the Lean prover as a comment block. The fine-tuned model (see Section 3.1) has been trained to recognize and utilize this format, using the incomplete code augmented with tactic state comments to guide subsequent proof generation.

## 4.2 Interactive Theorem Proving via Monte-Carlo Tree Search

Our proof search tree is developed using the standard Monte-Carlo Tree Search (MCTS) paradigm (MCTS; Coulom, 2006; Browne et al., 2012), which iteratively applies four steps: *Selection*, *Expansion*, *Simulation*, and *Backpropagation*. We integrate the *Simulation* step into *Expansion* because our whole-proof generation model inherently performs a rollout from the expanded node. The detailed design of the algorithm workflow is as follows.

**Selection.** The selection step, *a.k.a.* the tree policy, starts from the root node and traverses downward to identify a promising node for expansion. The objective of this algorithmic step is to trade off between exploration and exploitation (Kocsis & Szepesvári, 2006). The tree policy at a tree node $s$ is computed by selecting the action that maximizes the value from the set of valid operations:

$$\text{TreePolicy}(s) = \underset{a \in \text{Children}(s) \cup \{\oslash\}}{\arg\max} Q_{\text{UCB}}(s, a), \tag{1}$$

where the action $a$ can be either moving to a child node, denoted by $a \in \text{Children}(s)$, or expanding the current node $s$, denoted by a special token $a = \oslash$. This approach uses a technique called *virtual node* (Wang et al., 2023b), which assigns each node an imaginary child to represent the selection of the current node $s$ for expansion. It enables the tree search agent to continually expand non-leaf nodes, as the action space is supported by a generative model whose output scope cannot be determined by a fixed number of trails. The value estimation $Q_{\text{UCB}}(s, a)$ of performing action $a$ on node $s$ is composed by two components:

$$\forall a \in \text{Children}(s) \cup \{\oslash\}, \quad Q_{\text{UCB}}(s, a) = \underbrace{Q(s, a)}_{\text{Exploitation}} + \underbrace{\text{UCB}(s, a)}_{\text{Exploration}}, \tag{2}$$

where $Q(s, a)$ denotes a sample-based estimation of action values derived from the selection history, functioning as the exploitation component that retrieves high-value candidates from previous trials. $\text{UCB}(s, a)$ denotes the exploration bonus computed by upper confidence bounds (UCB; Auer, 2002), which diminishes with the repeated execution of the state-action pair $(s, a)$. More specifically, $Q_{\text{UCB}}(s, a)$ stands for an optimistic estimation of $Q(s, a)$ and can serve as an upper bound with high probability. We defer the discussion of detailed settings of node values and UCB bonus to Section 4.3.

**Expansion.** The next step is invoking the proof generation model to expand the node nominated by the selection phase. Resuming the incomplete proof codes stored on the node designated for expansion, we perform whole-proof generation to propose a series of subsequent tactics and submit the generated proof to Lean prover for verification. Such a trial of proof completion is equivalent to conducting a single rollout of simulation within the standard MCTS framework. When the verification result indicates the proof is complete, the search procedure is ready to be terminated, having found a new proof of the desired theorem. Otherwise, we parse the verification feedback and truncate the generated proof to the assertion of the earliest verification error. The remaining tactics are transformed into a path of nodes to be merged into the search tree (see Figure 2). It is important to note that, because we use the whole-proof generation setting—where the output is an entire proof consisting of a sequence of tactics, rather than just the next tactic—our expansion procedure may insert a path of tree nodes into the search tree during each iteration. This differs from the conventional MCTS designed for competitive games, which typically expands only one layer of children nodes per iteration (Silver et al., 2016; 2018; Schrittwieser et al., 2020).

**Backpropagation.** The final phase of each tree search iteration is to update value statistics along the selection trajectory from the root to the expanded node, *i.e.*, updating the values associated with the tree policy stated in Eq. (1). Let $\tau = \{(root, s^{(1)}), (s^{(1)}, s^{(2)}), (s^{(2)}, s^{(3)}), \ldots, (s^{(|\tau|-1)} = s_t, \oslash)\}$ denote the selection trajectory of $t$-th iteration that ends with $s_t$ as the expanding node. We update $Q_{\text{UCB}}(s, a)$ for all $(s, a) \in \tau$ by taking the most recent trajectory reward $R(\tau)$ into account (details refer to Eq. (7)). The extrinsic source of rewards comes from the compiler feedback, specifically assigning a reward of $R_{\text{extrinsic}}(\tau) = 1$ for completed proofs and $R_{\text{extrinsic}}(\tau) = 0$ for unsolved ones. In Section 4.3, we will introduce an intrinsic reward mechanism to augment the reward assignment that enhances the agent's incentive for exploration.

### 4.3 INTRINSIC REWARDS FOR PROOF SEARCH

In the search problem of formal theorem proving, the extrinsic rewards are extremely sparse, *i.e.*, the search agent only obtains non-zero rewards when the proof is completely solved. More specifically, the proof search process forms a tree structure with only a narrow set of leaves delivering non-zero rewards, which matches a famous hard-exploration case (Krishnamurthy et al., 2016) in the literature of statistical reinforcement learning. To promote exploration in sparse-reward sequential decision making, one classical paradigm is constructing intrinsic rewards (Schmidhuber, 2010) that encourage the agent to not only optimize extrinsic rewards but also acquire general information about the interactive environment (Bellemare et al., 2016; Houthooft et al., 2016; Pathak et al., 2017; Burda et al., 2019). In this section, we present our intrinsic-reward-driven exploration algorithm, *RMax applied to Tree Search* (RMaxTS), to incorporate reward-free exploration in the proof search problem.

**RMax applied to MCTS.** We adopt RMax (Brafman & Tennenholtz, 2002), a classical exploration mechanism, to construct intrinsic rewards for Monte-Carlo tree search. The core idea of RMax is to explore a broad coverage of the state space. The agent awards itself a maximal amount of reward upon reaching an unseen state. In the context of proof search, where no extrinsic rewards are provided until the proof is completed, our algorithmic procedure resembles ZeroRMax (Jin et al., 2020), in which the agent's exploration is driven solely by intrinsic rewards, *i.e.*, setting $R(\tau) = R_{\text{intrinsic}}(\tau)$. The intrinsic reward of a tree expansion step is determined by whether a new node is added to the search tree,

$$R_{\text{intrinsic}}(\tau) = \mathbb{I}\left[\text{at least one new node is added to the search tree}\right], \tag{3}$$

where $\tau$ denotes the most recent selection trajectory that requires a reward assignment for back-propagation. This exploration strategy prioritizes the expansion of nodes where the prover model generates tactics that lead to a diverse range of tactic states. As multiple Lean codes can result in the same transition of intermediate states, this heuristics can potentially reduce redundant generation and improve sample efficiency.

**UCB for Non-stationary Rewards.** The common setting of UCB exploration bonus for Monte-Carlo tree search is using UCB1 (Auer et al., 2002):

$$Q_{\text{UCB1}}(s, a) = \frac{W(s, a)}{N(s, a)} + \sqrt{\frac{2 \ln \sum_{a'} N(s, a')}{N(s, a)}}, \tag{4}$$

$$W(s, a) = \sum_{\tau \in \Gamma(s, a)} R(\tau), \tag{5}$$

$$N(s, a) = |\Gamma(s, a)|, \tag{6}$$

where $\Gamma(s, a) = \{\tau \mid (s, a) \in \tau\}$ denotes the list of tree-policy trajectory $\tau$ containing $(s, a)$ as an intermediate selection step. To facilitate discussions, we organize the list $\Gamma(s, a) = \{\tau_1, \tau_2, \cdots\}$ such that newly collected trajectories have larger subscript indices. In this work, we propose to use an alternative variant of UCB method. Note that the derived intrinsic reward in Eq. (3) is a non-stationary reward signal whose expected value decays with the progress of exploration. That is because it becomes definitely harder to discover new nodes with unseen tactic states as the search tree expands through sophisticated exploration. To tackle the non-stationarity, we consider *discounted upper confidence bounds* (DUCB; Garivier & Moulines, 2011), which uses a discount factor $\gamma \in (0, 1)$ to smoothly drop those outdated feedback records:

$$Q_{\text{DUCB}}(s, a) = \frac{W_\gamma(s, a)}{N_\gamma(s, a)} + \sqrt{\frac{2 \ln \sum_{a'} N_\gamma(s, a')}{N_\gamma(s, a)}}, \tag{7}$$

$$W_\gamma(s, a) = \sum_{t=1}^{N(s, a)} \gamma^{N(s, a) - t} R(\tau_t), \tag{8}$$

$$N_\gamma(s, a) = \sum_{t=0}^{N(s, a) - 1} \gamma^t, \tag{9}$$

where newly received feedback would be assigned a larger weight in the value estimation. In practice, we set $\gamma = 0.99$. Note that the role of discount factor $\gamma$ in DUCB differs from its role in value iteration for infinite-horizon MDPs. The discounting is applied to tree search iterations rather than to the action-step horizon within a single trajectory.

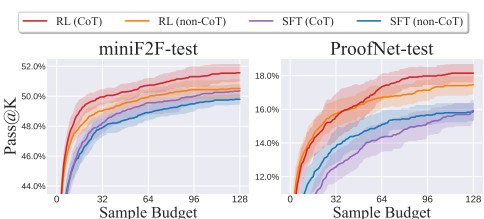

| Model | Pass@128 | |
| --- | --- | --- |
| | miniF2F-test | ProofNet-test |
| Base (3-shot) | $29.7\% \pm 0.5\%$ | $9.7\% \pm 0.7\%$ |
| SFT (non-CoT) | $49.8\% \pm 0.3\%$ | $15.9\% \pm 0.5\%$ |
| SFT (CoT) | $50.4\% \pm 0.4\%$ | $15.9\% \pm 0.6\%$ |
| RL (non-CoT) | $50.5\% \pm 0.6\%$ | $17.5\% \pm 0.5\%$ |
| RL (CoT) | $51.6\% \pm 0.5\%$ | $18.2\% \pm 0.5\%$ |

Figure 3: **Comparison of model capabilities at different training stages.** "CoT" and "non-CoT" refer to evaluations using two guiding prompts. The shaded region represents the range of standard deviations around the mean values. The notation $\mu \pm \sigma$ indicates the average accuracy $\mu$ and the standard deviation $\sigma$, estimated from 16 independent runs of sampling evaluation.

## 5 EXPERIMENTAL RESULTS

In this section, we evaluate the theorem-proving capabilities of DeepSeek-Prover-V1.5 using two distinct benchmarks: miniF2F (Zheng et al., 2022), which encompasses high-school level exercises and competition problems, and ProofNet (Azerbayev et al., 2023), which pertains to undergraduate-level theorems. We present the results for both whole-proof generation and Monte-Carlo tree search methodologies. Detailed experiment settings are described in Appendix B.1.

### 5.1 MAIN RESULTS

**Results on miniF2F and ProofNet.** Table 1 and 2 provides a comparative analysis of various theorem-proving methods on the miniF2F and ProofNet benchmarks. In the single-pass whole-proof generation setting, DeepSeek-Prover-V1.5-RL achieved the highest pass rate at 60.2% on miniF2F-test and at 23.7% on ProofNet-test, significantly outperforming all advanced baselines. When combining DeepSeek-Prover-V1.5-RL with RMaxTS, the new state-of-the-art are achieved, solving 62.7% problems from miniF2F-test and 25.3% problems from ProofNet-test.

**General Enhancement of Reinforcement Learning.** To support the claim that online reinforcement learning from verification feedback generally enhances the model capabilities, we compare our final model to the SFT-only version using a large sample budget. The comparison results are presented as two columns in Table 4. DeepSeek-Prover-V1.5-RL consistently outperforms the SFT model across all generation settings, regardless of whether the chain-of-thought strategy is applied. The results also indicate that the improvements gained from conducting online RL is orthogonal to those achieved through RMaxTS, which can be further combined to boost the performance. By integrating both CoT prompting and RMaxTS, DeepSeek-Prover-V1.5-RL achieves a pass rate of 62.7% on miniF2F-test. This performance shows a notable 3.7% improvement over the SFT model, highlighting the critical role of reinforcement learning in enhancing the overall effectiveness of the proof completion model.

**CoT, non-CoT, and Mixture Strategy.** We compare the performance of two generation modes, *i.e.*, non-CoT and CoT, on miniF2F-test dataset. The results, shown in Table 4, indicate that the advantage of CoT over the non-CoT mode is amplified as the sample budget increases. This suggests that the incorporation of natural language chain-of-thought can diversify the planning pathways of theorem proving, potentially leading to a broader range of reasoning strategies and more innovative solutions. Results also show that these two modes have complementary advantages across different problems. The model's theorem proving strategy in the CoT mode is more systematic and proactive in mathematical thinking, while in the non-CoT mode, the model can efficiently use Lean high-level tactics to solve computational problems that can be addressed within Lean's automation mechanisms. To leverage these advantages, we consider a mixture strategy, denoted by non-CoT & CoT in Table 4, allocates half of sample budget to the CoT mode and the remains to the non-CoT mode. This simple combination of two guiding prompts shows great promise in further bootstrapping the performance of our proof completion model, achieving a pass rate of 63.5% on miniF2F-test. In Appendix H, we present example problems that illustrate the different advantages of the two generation modes.

| Method | Sample budget | miniF2F-test |
|---|---|---|
| *Single-pass Whole-Proof Generation Methods* | | |
| DeepSeek-Prover-V1 (Xin et al., 2024) | 128 | $46.1\% \pm 0.5\%$ |
| | $16 \times 4096$ | $50.0\%$ |
| DeepSeek-Prover-V1.5-SFT | 128 | $50.4\% \pm 0.4\%$ |
| | 3200 | $53.3\% \pm 0.5\%$ |
| | $16 \times 6400$ | $57.4\%$ |
| DeepSeek-Prover-V1.5-RL | 128 | $51.6\% \pm 0.5\%$ |
| | 3200 | $54.9\% \pm 0.7\%$ |
| | $16 \times 6400$ | $\mathbf{60.2}\%$ |
| *Tree Search Methods* | | |
| GPT-f (Polu et al., 2022) | $64 \times 8 \times 512$ | $36.6\%$ |
| Hypertree Proof Search (Lample et al., 2022) | $64 \times 5000$ | $41.0\%$ |
| Lean-STaR (Lin et al., 2024) | $64 \times 1 \times 50$ | $46.3\%$ |
| InternLM2-Math-Plus-7B (Ying et al., 2024b) | $1 \times 32 \times 100$ | $43.4\%$ |
| InternLM2-StepProver (Wu et al., 2024) | $1 \times 32 \times 100$ | $48.8\%$ |
| | $64 \times 32 \times 100$ | $54.5\%$ |
| DeepSeek-Prover-V1.5-SFT + RMaxTS | $1 \times 3200$ | $53.5\% \pm 0.4\%$ |
| | $16 \times 6400$ | $59.0\%$ |
| | $32 \times 6400^{\dagger}$ | $60.2\%$ |
| DeepSeek-Prover-V1.5-RL + RMaxTS | $1 \times 3200$ | $55.0\% \pm 0.7\%$ |
| | $16 \times 6400$ | $\mathbf{62.7}\%$ |
| | $32 \times 6400^{\dagger}$ | $\mathbf{63.5}\%$ |

Table 1: Comparison with state-of-the-art methods on the miniF2F-test dataset. Unless otherwise specified, DeepSeek-Prover-V1.5-SFT and RL employ CoT mode prompting. The notation $\mu \pm \sigma$ indicates the average accuracy $\mu$ and the standard deviation $\sigma$. The symbol $\dagger$ indicates performance using a mixture strategy with two guiding prompts (see Section 5.1 for details). More baseline results are presented in Table 3 in Appendix. The detailed description of the evaluation setting is included in Appendix B.1.

| Method | Sample budget | ProofNet | | |
|---|---|---|---|---|
| | | valid$^{\ddagger}$ | test | all |
| *Single-pass Whole-Proof Generation Methods* | | | | |
| DeepSeek-Prover-V1.5-SFT | 128 | $19.9\% \pm 0.4\%$ | $15.9\% \pm 0.6\%$ | $17.9\% \pm 0.3\%$ |
| | 3200 | $20.7\% \pm 0.7\%$ | $21.0\% \pm 0.9\%$ | $20.9\% \pm 0.6\%$ |
| | $4 \times 6400$ | $22.2\%$ | $23.7\%$ | $22.9\%$ |
| DeepSeek-Prover-V1.5-RL | 128 | $20.1\% \pm 0.5\%$ | $18.2\% \pm 0.5\%$ | $19.1\% \pm 0.4\%$ |
| | 3200 | $21.4\% \pm 0.3\%$ | $22.0\% \pm 0.5\%$ | $21.7\% \pm 0.4\%$ |
| | $4 \times 6400$ | $21.6\%$ | $23.7\%$ | $22.6\%$ |
| *Tree Search Methods* | | | | |
| ReProver (Yang et al., 2023) | - | - | - | $13.8\%$ |
| InternLM2-StepProver (Wu et al., 2024) | $1 \times 32 \times 100$ | - | - | $18.1\%$ |
| DeepSeek-Prover-V1.5-SFT + RMaxTS | $1 \times 3200$ | $22.2\% \pm 0.7\%$ | $21.6\% \pm 0.2\%$ | $21.9\% \pm 0.4\%$ |
| | $4 \times 6400$ | $23.8\%$ | $\mathbf{25.8\%}$ | $24.8\%$ |
| DeepSeek-Prover-V1.5-RL + RMaxTS | $1 \times 3200$ | $22.0\% \pm 0.3\%$ | $21.5\% \pm 0.8\%$ | $21.8\% \pm 0.4\%$ |
| | $4 \times 6400$ | $25.4\%$ | $\mathbf{25.3\%}$ | $25.3\%$ |

Table 2: Comparing with state-of-the-arts on the ProofNet dataset. The notation $\mu \pm \sigma$ indicates the average accuracy $\mu$ and the standard deviation $\sigma$. $^{\ddagger}$ Note that the validation set of ProofNet is used to perform expert iteration in supervised fine-tuning.

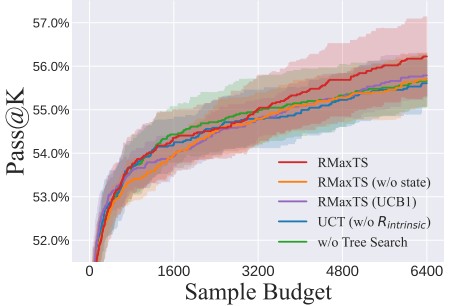

| | Sample budget | miniF2F-test |
|---|---|---|
| Single-Pass Generation | $4 \times 6400$ | $58.4\% \pm 0.5\%$ |
| | $16 \times 6400$ | $60.2\%$ |
| UCT (without $R_{\text{intrinsic}}$) | $4 \times 6400$ | $58.2\% \pm 0.3\%$ |
| | $16 \times 6400$ | $61.1\%$ |
| RMaxTS (DUCB $\rightarrow$ UCB1) | $4 \times 6400$ | $58.6\% \pm 0.3\%$ |
| | $16 \times 6400$ | $60.7\%$ |
| RMaxTS (without tactic state) | $4 \times 6400$ | $58.4\% \pm 0.3\%$ |
| | $16 \times 6400$ | $61.1\%$ |
| RMaxTS | $4 \times 6400$ | $59.6\% \pm 0.6\%$ |
| | $16 \times 6400$ | $62.7\%$ |

Figure 4: A modular ablation study on the design of RMaxTS. The experiments are conducted on the miniF2F-test dataset with DeepSeek-Prover-V1.5-RL using the CoT mode. The left panel presents the curves of Pass@K accuracy within 6400 generation samples. The results with a larger sample size are presented in the right panel. The shaded regions represent one standard deviation around the mean accuracy. The notation $\mu \pm \sigma$ indicates the average accuracy $\mu$ and the standard deviation $\sigma$.

## 5.2 ABLATION STUDIES ON RMAXTS

**Intrinsic Rewards and Discounted UCB.** We investigate the effectiveness of two core components of RMaxTS, *i.e.*, the intrinsic rewards defined in Eq. (3) and the discounted upper confidence bound stated in Eq. (7). We start with a baseline implementing the standard UCT algorithm (Kocsis & Szepesvári, 2006) without intrinsic rewards, in which the exploration is driven exclusively by the UCB bonus. Note that, since no non-zero rewards are provided for this baseline, all variants of the UCB formula become equivalent, as node selection is determined solely by visitation counts. The experimental results in Figure 4 show that, in the absence of intrinsic rewards, the performance of UCT (without $R_{\text{intrinsic}}$) degenerates into a level comparable to that of non-search methods. Furthermore, we consider RMaxTS using the standard UCB1 (refer to Eq. (4)) instead of the discounted UCB, denoted by RMaxTS (DUCB $\rightarrow$ UCB1). The results indicate that the performance of RMaxTS with UCB1 bonus is also moderate, comparable to that of UCT (without $R_{\text{intrinsic}}$). That is because UCB1 is designed to guarantee asymptotic performance through exhausted exploration (Auer et al., 2002) assuming the sample size to be sufficiently large. In contrast, the discounted UCB can accelerate the value propagation of non-stationary intrinsic rewards.

**Guidance of Tactic State Information.** When expanding a tree node, we concatenate the intermediate tactic state information as a comment block to the incomplete code to guide the proof completion. With the provided auxiliary information, the proof completion model can enhance its internal representation of the tactic state, offering intermediate guidance for long-horizon planning. To demonstrate this advantage, we present experiments on RMaxTS that performs code completion directly from the raw incomplete code without accessing tactic state information, denoted by RMaxTS (without tactic state) in Figure 4. The results indicate that the performance gain from applying tree search becomes moderate in the absence of tactic state information, especially when tackling hard problems that require a large amount of samples.

## 6 CONCLUSION

The framework of DeepSeek-Prover-V1.5 is designed to establish an AlphaZero-like pipeline for formal theorem proving. The use of expert iteration and synthetic data mirrors the core trial-and-error loop of reinforcement learning, with the compiler oracle serving as the world model to provide environmental supervision. Within the RL paradigm, the integrated tree search module has proven to be highly effective in advancing superhuman performance across various domains (Silver et al., 2016; Fawzi et al., 2022; Lutz et al., 2023). A promising future direction is training a critic model to assess incomplete proofs and prune search branches of proofs. Such a partial-proof critic model would implicitly perform temporal credit assignment (Sutton, 1984), decomposing proof-level feedback into step-wise value differences (Arjona-Medina et al., 2019). Developing critic models for assessing long planning paths and providing guidance rewards presents a crucial and challenging problem (Ng & Russell, 2000; Sorg et al., 2010) that warrants further investigation.

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

## A  IMPLEMENTATION DETAILS

### A.1  PRE-TRAINING

To enhance our language model's proficiency in generating formal proofs and reasoning through mathematical language, we further pre-train our base model from DeepSeekMath-7B-Base (Shao et al., 2024). This refinement involved training on high-quality datasets that include both code and natural language mathematical content. We specifically focused on formal languages widely used in proof assistants, such as Lean, Isabelle, and Metamath. We designate this improved model as DeepSeek-Prover-V1.5-Base.

**Potential Data Leak.** When constructing our SFT dataset, we ensure that all Lean statements in the training set have distinct initial states from those in the test set. The most uncontrollable potential source of data leakage stems from the pre-training phase. To the best of our knowledge, the leakage risk of Lean statement from miniF2F and ProofNet is limited, as these benchmarks were specifically designed for ML research and were not gathered from arbitrary sources on the internet. On the other hand, potential leakage of natural language statements from math problems may exist, but it is generally considered acceptable, as the formalization of natural language proof to Lean codes is one of the core challenges of formal theorem proving.

## A.2 SUPERVISED FINE-TUNING

**Expert Iteration.** We develop a comprehensive Lean 4 code completion dataset for the supervised fine-tuning. This dataset includes synthetic proof code derived from a wide range of formal theorems. These theorems are sourced from various projects, such as the standard Lean 4 math library Mathlib4 (Mathlib Community, 2020), synthetic theorems from DeepSeek-Prover-V1 (Xin et al., 2024) and Lean Workbook (Ying et al., 2024a), and validation sets from the miniF2F (Zheng et al., 2022) and ProofNet (Azerbayev et al., 2023) benchmarks. To augment the formal proof data, we employed an expert iteration process (Polu & Sutskever, 2020). This involves generating proofs using the language model, verifying the generated proof data, retraining the model with the verified data, and then using the optimized model to generate additional proof data. Between each iteration, we use DeepSeek-Coder V2 236B (Zhu et al., 2024) to annotate the thought process before the proof code as comments. We then tailor these data for the truncate-and-resume mechanism for RMaxTS (details in Section 4.1). The validated proofs generated through tree search are gathered to perform the next iteration of SFT. The final proof dataset consists of 9,645k sequences.

**Training Setting.** We conduct supervised fine-tuning based on the pre-trained model and train for 9B tokens, using a batch size of 2,048 and a constant learning rate of 1e-4. The training process begins with 100 warm-up steps to stabilize the learning dynamics. Training examples are randomly concatenated to form sequences, with a maximum context length of 4,096 tokens. The entire SFT phase processes 2600 steps and around 20B tokens.

## A.3 REINFORCEMENT LEARNING FROM PROOF ASSISTANT FEEDBACK

**Prompts.** In the reinforcement learning stage, we use a subset of theorem statements from the supervised fine-tuning dataset as training prompts. We select theorems for which DeepSeek-Prover-V1.5-SFT has a moderate success rate in generating correct proofs upon multiple attempts. This ensures that the model has room for improvement while still being able to receive positive feedback. After filtering, we retain approximately 4.5k unique theorem statements. Each theorem is prefixed with both CoT and non-CoT guiding prompts to enhance the model's proof generation capabilities in both modes.

**Rewards.** When training LLMs via RL, a trained reward model typically provides feedback signals. In contrast, formal theorem proving benefits from the rigorous verification of generated proofs by proof assistants, offering a significant advantage. Specifically, each generated proof receives a reward of 1 if verified as correct, and 0 otherwise. While this binary reward signal is accurate, it is also sparse, especially for theorems that are challenging for the supervised fine-tuned model. To mitigate this sparsity, we select training prompts that are challenging yet achievable for the supervised fine-tuned model, as described above.

**Reinforcement Learning Algorithm.** We employ the Group Relative Policy Optimization (GRPO; Shao et al., 2024) as our RL algorithm, which has demonstrated superior effectiveness and efficiency compared to PPO (Schulman et al., 2017), primarily because it eliminates the necessity of training an additional critic model. Specifically, GRPO samples a group of candidate proofs for each theorem prompt and optimizes the model based on the relative rewards of the outputs within the group. Given a group of outputs $\{o_1, \cdots, o_G\}$ regarding an input question $q$, the objective of

GRPO is stated as follows:

$$J_{\text{GRPO}}(\theta) = \frac{1}{G} \sum_{i=1}^{G} \frac{1}{|o_i|} \sum_{t=1}^{|o_i|} \Big[ L_{\text{clip}}(o_i, \theta) - \beta \mathbb{D}_{\text{KL}} \big[ \pi_\theta \parallel \pi_{\theta_{\text{ref}}} \big] \Big],$$

$$L_{\text{clip}}(o_i, \theta) = \min \left( \frac{\pi_\theta(o_{i,t} \mid q, o_{i,<t})}{\pi_{\theta_{\text{old}}}(o_{i,t} \mid q, o_{i,<t})} \hat{A}_{i,t}, \ \text{clip} \left( \frac{\pi_\theta(o_{i,t} \mid q, o_{i,<t})}{\pi_{\theta_{\text{old}}}(o_{i,t} \mid q, o_{i,<t})} \hat{A}_{i,t}, 1 - \epsilon, 1 + \epsilon \right) \right),$$

where $\hat{A}_{i,t} = \frac{r_i - \text{mean}(r)}{\text{std}(r)}$ is the normalized advantage with respect to the sampling-average baseline. Our prompt selection strategy is designed to likely include both correct and incorrect proofs among the candidates, aligning well with the group-relative nature of GRPO and thereby enhancing the training process.

**Training Setting.** We conduct RL training based on the SFT model, which serves as both the initial model and the reference model for imposing the Kullback-Leibler (KL) divergence penalty. We use a constant learning rate of 5e-6, and the KL penalty coefficient is set to 0.02. For each theorem, we sample a group of 32 candidate proofs, with maximum length set to 2,048. The training batch size is configured to 512. The entire RL phase processes 800 steps and around 1.5B tokens.

## A.4 Parallelization of Monte-Carlo Tree Search

To enhance the efficiency of Monte-Carlo Tree Search (MCTS), we implement several established parallelization techniques as described by Chaslot et al. (2008).

- **Root Parallelization:** We deploy 256 MCTS runners per node, with one language model per GPU and a batch size of 512 for proof generation. The Lean prover is invoked through REPL and executed on a cluster with thousands of CPU cores, where each proof verification task is handled by an individual process, created and terminated in a sandbox. Both proof generation by language models and verification by Lean provers are handled asynchronously. This setup allows MCTS runners to perform concurrent tree search operations, significantly accelerating the process.

- **Tree Parallelization:** We manage each search tree with 32 thread workers to parallelize the tree iteration steps. This method effectively schedules and balances the tasks of proof generation and Lean verification. Each thread worker iteratively performs the tree search loop by selecting a candidate node for expansion, invoking the language model to generate the proof, verifying the generated proof with the Lean prover, and performing backpropagation.

- **Virtual Loss:** To encourage diverse node selection among concurrent thread workers, we assign a virtual reward $R(\tau) = 0$ for ongoing iterations. This involves backpropagating a reward of $0$ temporarily and updating it to the true reward upon completion. This strategy promotes exploration of different nodes for expansion, thereby enhancing the overall search efficiency.

A.5  PSEUDO CODE OF RMAXTS

---
**Algorithm 1** RMax applied to Tree Search (RMaxTS)
---
1: **function** MCTS($root$, $sample\_budget$)
2:     **for** $t = 1$ to $sample\_budget$ **do**
3:         $node \leftarrow$ SELECT($root$)
4:         $node' \leftarrow$ EXPAND($node, t$)
5:         $reward \leftarrow \mathbb{I}[node'.time\_stamp \neq t]$         ▷ RMax intrinsic reward (Eq. (3))
6:         BACKPROPAGATE($node, reward$)
7:     **end for**
8:     **return** BESTCHILD($root$)
9: **end function**

10: **function** SELECT($node$)
11:     $\mathcal{A} \leftarrow$ Children($node$) $\cup \{\oslash\}$         ▷ $\oslash$ denotes the virtual node
12:     $child \leftarrow \arg\max_{a \in \mathcal{A}} Q_{\text{DUCB}}(node, a)$         ▷ Discounted UCB (Eq. (7))

$$\text{where}\ \ Q_{\text{DUCB}}(node, a) = \frac{W_\gamma(node, a)}{N_\gamma(node, a)} + \sqrt{\frac{2\ln\sum_{a' \in \mathcal{A}} N_\gamma(node, a')}{N_\gamma(node, a)}}$$

13:     **if** $child = \oslash$ **then**
14:         **return** $node$         ▷ select the virtual node
15:     **else**
16:         **return** SELECT($child$)
17:     **end if**
18: **end function**

19: **function** EXPAND($node, t$)
20:     Resume the partial code stored in $node$
21:     Generate a new proof through code completion
22:     Truncate the generated proof to the first compilation error
23:     Segment the valid code into a series of state transitions ($node.state \rightarrow s_1 \rightarrow \cdots \rightarrow s_K$)
24:     **for** $k = 1$ to $K$ **do**
25:         **if** $node$ has a child with state $s_k$ **then**         ▷ move to an existing child
26:             $node \leftarrow child$ where $child \in$ Children($node$) $\cap\ child.state = s_k$
27:         **else**         ▷ create a new tree node
28:             create a new tree node $node'$
29:             $node'.state \leftarrow s_k$
30:             $node'.time\_stamp \leftarrow t$
31:             Children($node$) $\leftarrow$ Children($node$) $\cup \{node'\}$
32:             $node \leftarrow node'$
33:         **end if**
34:     **end for**
35:     **return** $node$
36: **end function**

37: **function** BACKPROPAGATE($node, reward$)
38:     $N_\gamma(node, \oslash) \leftarrow \gamma \cdot N_\gamma(node, \oslash) + 1$         ▷ update the virtual node
39:     $W_\gamma(node, \oslash) \leftarrow \gamma \cdot W_\gamma(node, \oslash) + reward$
40:     **while** $node.parent$ is not null **do**
41:         $child \leftarrow node$
42:         $node \leftarrow node.parent$
43:         $N_\gamma(node, child) \leftarrow \gamma \cdot N_\gamma(node, child) + 1$         ▷ update the tree path
44:         $W_\gamma(node, child) \leftarrow \gamma \cdot W_\gamma(node, child) + reward$
45:     **end while**
46: **end function**
---

### A.6 COMPARISON WITH EXISTING METHODS

In this section, we compare our proposed proof tree search method, which introduces a novel truncate-and-resume mechanism for whole-proof generation, with existing approaches. Current methods for using language models in formal mathematics proof search generally fall into two main strategies:

- **Multi-pass proof-step generation**: This strategy breaks down the proving process into multiple episodes of tactic generation and verification, typically following a **tree search** pattern. It involves generating and verifying one tactic at a time, repeating the process for the next tactic until no proof goals remain. Notable examples include GPT-f (Polu & Sutskever, 2020; Polu et al., 2022), Thor (Jiang et al., 2022a), ReProver (Yang et al., 2023), Hypertree Proof Search (Lample et al., 2022), and InternLM2-StepProver (Wu et al., 2024).

- **Single-pass whole-proof generation**: This approach generates and verify an entire proof in one attempt. If the proof is incorrect, the model generates a new proof in the next attempt. Methods in this category include DSP (Jiang et al., 2022b), Subgoal-Prover Zhao et al. (2023), LEGO-Prover (Wang et al., 2023a), Lyra (Zheng et al., 2023), and miniCTX (Hu et al., 2024).

Our proof tree search method uniquely bridges these two strategies, offering a novel hybrid approach. It starts with whole-proof generation, similar to the single-pass approach, but extends this by implementing a sophisticated truncate-and-resume mechanism. This process involves truncating the generated proof to its successful initial segment, parsing this segment into individual tactics, and resuming the tree search from this point. This iterative process effectively implements a Monte-Carlo Tree Search, seamlessly integrating single-pass whole-proof generation with multi-pass proof-step generation. Consequently, we can train a single model with nearly identical objectives to support both strategies simultaneously. Our experimental results demonstrate that this unified approach achieves superior performance in both settings. By combining the strengths of existing methods and introducing innovative techniques, our method offers a more versatile and effective solution for formal mathematics proof search, potentially paving the way for future advancements in this field.

## B EXPERIMENT SETTINGS

### B.1 EVALUATION

**Formal Verification Environment.** The Lean theorem prover is a state-of-the-art proof assistant and functional programming language, primarily designed for formalizing mathematics. Developed by Leonardo de Moura at Microsoft Research (Moura & Ullrich, 2021), Lean combines dependent type theory with a versatile meta-programming framework, facilitating the creation of proofs and the development of domain-specific tactics. Its open-source nature, coupled with an extensive library ecosystem such as `mathlib` (Mathlib Community, 2020), has established Lean as a cornerstone of modern mathematical formalization (Avigad, 2023). Lean has hosted some of the world's most sophisticated formalizations, including scheme theory (Buzzard et al., 2021), forcing (Han & van Doorn, 2019), and the ongoing project for Fermat's Last Theorem (Buzzard, 2024).

A defining feature of Lean is its tactic-based framework, which enables the construction of proofs through high-level, human-readable commands that automate common reasoning patterns. Tactics allow users to decompose complex goals into simpler subgoals and guide the proof process interactively. Moreover, Lean's extensibility supports the creation of custom tactics tailored to specific domains, offering significant optimizations and abstractions. This tactic-based approach bridges the gap between the machine's precision and human intuition, making Lean exceptionally effective for large-scale formalizations and interactive theorem proving.

Below is an example that demonstrates the application of tactics in Lean 4 and their corresponding tactic states. First, we declare a theorem statement to prove:

```
import Mathlib

theorem Add_Assoc (a b c : Nat) : a + (b + c) = (a + b) + c := by sorry
```

Here, `sorry` serves as a placeholder for the actual proof. Lean provides the initial tactic state as feedback:

```
a b c : ℕ
⊢ a + (b + c) = a + b + c
```

To complete the proof, we can replace it with tactics like `ring` or `rw [Nat.add_assoc]` to resolve the goal. After applying one of these tactics, Lean responds with `No goals`, indicating that all objectives have been satisfied.

In this work, the transitions between tactic states are traced using the REPL (Leanprover Community, 2023), and these transitions are used as inputs for training language models. Notably, the transitions between tactic states employed in our proof search are shared across all tactic-based proof assistants, including Isabelle (Paulson, 1994) and Coq (Coq Development Team, 2024). Consequently, there are no theoretical barriers to adapting our methods for use with other proof assistants.

**Benchmarks.** We evaluate theorem-proving performance on the following benchmarks to compare model capabilities after each training stage:

- **MiniF2F** (Zheng et al., 2022) focuses on formal problem-solving skills for high-school level exercises and competitions, such as AMC, AIME, and IMO, with an emphasis on algebra and number theory. The benchmark includes 244 validation and 244 test problems, originally in Lean 3 and manually converted to Lean 4.9.0, based on the version provided by Yang (2023).
- **ProofNet** (Azerbayev et al., 2023) evaluates formal theorem-proving capabilities at the undergraduate level in mathematics. It comprises 185 validation and 186 test problems from widely-used undergraduate textbooks, covering real and complex analysis, linear algebra, abstract algebra, and topology. These problems were initially in Lean 3 and manually converted to Lean 4.9.0.

**Prompting Configurations.** For each proof attempt of DeepSeek-Prover-V1.5-Base, we independently sample three proof demonstrations from the validation set to construct the few-shot prompts. For the miniF2F benchmark, we use human-written proofs from Yang (2023), while for the ProofNet benchmark, we use correct proofs generated by DeepSeek-Prover-V1.5-RL as few-shot demonstrations. For DeepSeek-Prover-V1.5-SFT and DeepSeek-Prover-V1.5-RL, we employ two types of guiding prompts: one that encourages chain-of-thought (CoT) reasoning before each proof step, and one that does not (non-CoT). Detailed examples are provided in Appendix G.

**Evaluation Settings.** We evaluate theorem-proving performance using the pass@$K$ accuracy metric, which measures the model's success in generating a correct proof within $K$ attempts. Each model is deployed on a single A100-40G GPU, utilizing the vLLM framework (Kwon et al., 2023) for sample generation. The sampling parameters are set with a temperature of 1, a top-p value of 0.95, and a maximum token limit of 2,048. The generated proofs are then verified using the Lean 4 theorem prover. For this verification, we import Mathlib4 (Mathlib Community, 2020) and Aesop (Limperg & From, 2023) to access predefined premises and tactics. The verification process is subject to a time limit of 300 seconds. The largest-scale experiment presented in this paper, RMaxTSwith a $32 \times 6400$ sample size and CoT generation mode on the miniF2F-test dataset, requires 48 A100 GPUs and approximately 2000 CPUs to complete within 48 hours.

**Baselines.** We present a comparative analysis of DeepSeek-Prover-V1.5 against previous state-of-the-art language models, highlighting its performance and advancements.

- **General-purpose Models: GPT-3.5** and **GPT-4** (OpenAI, 2023) are advanced generative AI models developed by OpenAI, known for their effectiveness across diverse tasks, including code generation. Despite not being specifically designed for theorem proving, their extensive parameter scales provide significant capabilities. The evaluation of these models in formal theorem proving is facilitated by **COPRA** (Thakur et al., 2023), an in-context learning agent that leverages these large language models to propose tactic applications. Additionally, we examine **Llemma** (Azerbayev et al., 2024), a series of language models

trained on extensive general mathematical corpora, commonly used as the base model for formal theorem proving.

- **Specialized Models for Formal Mathematics: GPT-f** (Polu & Sutskever, 2020; Polu et al., 2022) represents an initial effort to apply Transformers (Vaswani et al., 2017) to proof-step generation for theorem proving tasks, utilizing a best-first search module to construct complete proofs. Subsequent advancements include **ReProver** (Yang et al., 2023), **LLMStep** (Welleck & Saha, 2023), and **Lean-STaR** (Lin et al., 2024). **Hypertree Proof Search** (Lample et al., 2022) explores the use of Monte Carlo tree search in formal theorem proving using Lean. Concurrent works, **InternLM2-Math** (Ying et al., 2024b) and **InternLM2-StepProver** (Wu et al., 2024), also demonstrate outstanding performance.

**Metric.** We compare the performance of DeepSeek-Prover-V1.5 with state-of-the-art models using the pass@$K$ accuracy metric, which evaluates the model's ability to generate a correct proof within $K$ attempts. We display the sample budget $K$ according to the the following rules to align the computation budget across different generation schemes.

- For single-pass sampling methods, we define the sample budget $K$ as the total number of proofs generated, with large values of $K$ factorized for the ease of comparison to tree search methods.
- For best-first-search methods, following the notation of Azerbayev et al. (2024), we present $K = N \times S \times T$ where $N$ denotes the number of best-first-search attempts, $S$ denotes the number of tactics generated for each expansion, and $T$ denotes the number of expansion iterations.
- For tree search methods, *e.g.*, RMaxTS and HTPS (Lample et al., 2022), we present $K = N \times T$ where $N$ denotes the number of tree search attempts, and $T$ denotes the number of model generations invoked in tree expansions.

**Confidence Interval.** We report pass@$K$ accuracy as $\mu \pm \sigma$ where $\mu$ indicates the average accuracy and $\sigma$ denotes the standard deviation. The average score and the standard deviation are computed by running several independent trials (using different random seeds) to replicate the sampling and evaluation process. Let $\{a_i\}_{i=1}^n$ denote the evaluation results of $n$ independent runs. The average accuracy is calculated by $\mu = \frac{1}{n} \sum_{i=1}^n a_i$. The standard deviation is calculated by $\sigma = \sqrt{\frac{1}{n} \sum_{i=1}^n (a_i - \mu)^2}$. For the miniF2F dataset, experiments with a sample budget below $4 \times 6400$ are based on 16 independent runs, while those with a budget of exactly $4 \times 6400$ are estimated from 4 independent runs. For the ProofNet dataset, the threshold is set at 3200. The results of ProofNet with a sample budget below 3200 are derived from 16 independent runs, while those with a sample budget of exactly 3200 are estimated from 4 independent runs.

| Dataset | Sample budget | #Runs | Dataset | Sample budget | #Runs |
|---------|---------------|-------|---------|---------------|-------|
| miniF2F | $< 4 \times 6400$ | 16 | ProofNet | $< 3200$ | 16 |
|         | $= 4 \times 6400$ | 4 |          | $= 3200$ | 4 |
|         | $> 4 \times 6400$ | 1 |          | $> 3200$ | 1 |

**Checkpoint Selection.** The final models are selected from three training runs using different random seeds. During each training run, the model is evaluated every 100 training steps, with the Pass@128 score calculated for the miniF2F-test. Due to computational resource constraints, the training-time evaluation is conducted using a single trial of 128 samples. We select the checkpoint that achieves peak performance over several consecutive steps as the candidate for a specific training run. Upon completing three training runs, we obtain three checkpoint candidates. We then perform evaluation using 16 independent trials to compute the mean and standard deviation. The RL training phase is performed on the selected SFT model. The results demonstrate that the effects of random seeds are marginal compared to the overall improvement achieved during the SFT and RL phases.

| minF2F-test | SFT phase (Pass@128) | | |
| --- | --- | --- | --- |
| | selected | seed-2 | seed-3 |
| CoT | $50.6\% \pm 0.5\%$ | $50.3\% \pm 0.3\%$ | $49.9\% \pm 0.4\%$ |
| non-CoT | $49.8\% \pm 0.4\%$ | $48.8\% \pm 0.4\%$ | $49.6\% \pm 0.3\%$ |
| minF2F-test | RL phase (Pass@128) | | |
| | selected | seed-2 | seed-3 |
| CoT | $51.4\% \pm 0.5\%$ | $51.2\% \pm 0.4\%$ | $51.2\% \pm 0.5\%$ |
| non-CoT | $50.5\% \pm 0.4\%$ | $50.7\% \pm 0.5\%$ | $50.2\% \pm 0.4\%$ |

To mitigate the risk of overestimating our evaluation results, we perform cross-validation to ensure the reported scores are robust and reliable. Specifically, we first conduct 2 (modes) $\times$ 3 (train seeds) $\times$ 16 (eval seeds) $\times$ 128 samples to compare the Pass@128 scores of three seed candidate. Upon selecting the final model, we perform an additional 2 (modes) $\times$ 16 (eval seeds) $\times$ 128 samples to recalculate the $\mu \pm \sigma$ scores for the chosen model. The recomputed scores are then reported in the main text. Note that the recomputed scores provide an unbiased evaluation of the chosen model, as the evaluation is conducted independently from the selection process. This additional evaluation round is crucial to ensure the reported results are rigorous, as maximizing over random variables inherently leads to an overestimation of expected values.

## B.2 SUPPLEMENTARY EXPERIMENT RESULTS

| Method | Model size | Sample budget | miniF2F-test |
|---|---|---|---|
| *Single-pass Whole-Proof Generation Methods* | | | |
| TheoremLlama [64] | 8B | 128 | 33.6% |
| DeepSeek-Prover-V1 [68] | 7B | 128 | $46.1\% \pm 0.5\%$ |
| | | $16 \times 4096$ | 50.0% |
| DeepSeek-Prover-V1.5-Base | 7B | 128 | $29.7\% \pm 0.5\%$ |
| | | 3200 | 39.2% |
| | | 6400 | 42.2% |
| DeepSeek-Prover-V1.5-SFT | 7B | 32 | $48.2\% \pm 0.6\%$ |
| | | 64 | $49.6\% \pm 0.7\%$ |
| | | 128 | $50.4\% \pm 0.4\%$ |
| | | 3200 | $53.3\% \pm 0.5\%$ |
| | | $4 \times 6400$ | $55.8\% \pm 0.7\%$ |
| | | $16 \times 6400$ | 57.4% |
| DeepSeek-Prover-V1.5-RL | 7B | 32 | $50.0\% \pm 0.5\%$ |
| | | 64 | $50.7\% \pm 0.4\%$ |
| | | 128 | $51.6\% \pm 0.5\%$ |
| | | 3200 | $54.9\% \pm 0.7\%$ |
| | | $4 \times 6400$ | $58.4\% \pm 0.6\%$ |
| | | $16 \times 6400$ | **60.2**% |
| *Tree Search Methods* | | | |
| COPRA (Code Llama) [60] | | $1 \times 500$ | 5.7% |
| COPRA (GPT-3.5) [60] | | $1 \times 60$ | 9.0% |
| COPRA (GPT-4) [60] | | $1 \times 60$ | 26.6% |
| Llemma-7B [8] | 7B | $1 \times 32 \times 100$ | 26.2% |
| Llemma-34B [8] | 34B | $1 \times 32 \times 100$ | 25.8% |
| ReProver [70] | 229M | - | 26.5% |
| LLMStep [66] | 2.8B | $1 \times 32 \times 100$ | 27.9% |
| GPT-f [48] | 770M | $64 \times 8 \times 512$ | 36.6% |
| Hypertree Proof Search [35] | 600M | $64 \times 5000$ | 41.0% |
| Lean-STaR [38] | 7B | $64 \times 1 \times 50$ | 46.3% |
| InternLM2-Math-7B [73] | 7B | $1 \times 32 \times 100$ | 30.3% |
| InternLM2-Math-Plus-7B [73] | 7B | $1 \times 32 \times 100$ | 43.4% |
| InternLM2-StepProver [67] | 7B | $1 \times 32 \times 100$ | 48.8% |
| | | $64 \times 32 \times 100$ | 54.5% |
| DeepSeek-Prover-V1.5-SFT + RMaxTS | 7B | $1 \times 3200$ | $53.5\% \pm 0.4\%$ |
| | | $4 \times 6400$ | $56.3\% \pm 0.3\%$ |
| | | $16 \times 6400$ | 59.0% |
| | | $32 \times 6400^{\dagger}$ | 60.2% |
| DeepSeek-Prover-V1.5-RL + RMaxTS | 7B | $1 \times 3200$ | $55.0\% \pm 0.7\%$ |
| | | $4 \times 6400$ | $59.6\% \pm 0.6\%$ |
| | | $16 \times 6400$ | **62.7**% |
| | | $32 \times 6400^{\dagger}$ | **63.5**% |

Table 3: Comparison with state-of-the-art methods on the miniF2F-test dataset. The notation $\mu \pm \sigma$ denotes the average accuracy $\mu$ and the standard deviation $\sigma$. Unless otherwise specified, DeepSeek-Prover-V1.5-Base results are based on 3-shot prompting, while DeepSeek-Prover-V1.5-SFT and RL employ CoT mode prompting. The symbol $\dagger$ indicates performance using a mixture strategy with two guiding prompts.

| | Prompt mode | Sample budget | DeepSeek-Prover-V1.5 | |
| | | | SFT | RL |
|---|---|---|---|---|
| Single-Pass Generation | non-CoT | $4 \times 6400$ | $54.7\% \pm 0.4\%$ | $56.5\% \pm 0.5\%$ |
| | | $16 \times 6400$ | $56.1\%$ | $57.4\%$ |
| | CoT | $4 \times 6400$ | $55.8\% \pm 0.7\%$ | $58.4\% \pm 0.5\%$ |
| | | $16 \times 6400$ | $57.4\%$ | $60.2\%$ |
| | non-CoT & CoT | $(2+2) \times 6400$ | $56.1\% \pm 0.8\%$ | $58.3\% \pm 0.6\%$ |
| | | $(8+8) \times 6400$ | $58.2\%$ | $60.7\%$ |
| | | $(16+16) \times 6400$ | $58.6\%$ | $61.1\%$ |
| RMaxTS | non-CoT | $4 \times 6400$ | $55.7\% \pm 0.6\%$ | $58.4\% \pm 0.6\%$ |
| | | $16 \times 6400$ | $57.8\%$ | $59.4\%$ |
| | CoT | $4 \times 6400$ | $56.3\% \pm 0.3\%$ | $59.6\% \pm 0.6\%$ |
| | | $16 \times 6400$ | $59.0\%$ | $62.7\%$ |
| | non-CoT & CoT | $(2+2) \times 6400$ | $56.1\% \pm 0.8\%$ | $60.0\% \pm 0.8\%$ |
| | | $(8+8) \times 6400$ | $59.0\%$ | $63.1\%$ |
| | | $(16+16) \times 6400$ | $60.2\%$ | **63.5%** |

Table 4: A large-scale ablation study to investigate the effectiveness of several algorithmic designs on model training. The results are evaluated on the miniF2F-test dataset. The notation $\mu \pm \sigma$ denotes the average accuracy $\mu$ and the standard deviation $\sigma$.

## C PROBLEM CATEGORIES ON MINIF2F BENCHMARK

| miniF2F-test single-pass generation | | DeepSeek-Prover-V1 $16 \times 4096$ | DeepSeek-Prover-V1.5-SFT | | DeepSeek-Prover-V1.5-RL | |
| | | | $4 \times 6400$ | $16 \times 6400$ | $4 \times 6400$ | $16 \times 6400$ |
|---|---|---|---|---|---|---|
| Olympiad | IMO | 1/20 | 2/20 | 2/20 | 2/20 | 2/20 |
| | AIME | 4/15 | 5/15 | 6/15 | 5/15 | 6/15 |
| | AMC | 12/45 | 13/45 | 14/45 | 15/45 | 16/45 |
| MATH | Algebra | 53/70 | 55/70 | 56/70 | 55/70 | 56/70 |
| | Number Theory | 45/60 | 47/60 | 49/60 | 49/60 | 51/60 |
| Custom | Algebra | 4/18 | 8/18 | 8/18 | 9/18 | 9/18 |
| | Number Theory | 1/8 | 2/8 | 2/8 | 4/8 | 4/8 |
| | Induction | 2/8 | 3/8 | 3/8 | 3/8 | 3/8 |

Table 5: Problems solved by DeepSeek-Prover-V1.5 on miniF2F-test, grouped by miniF2F categories. The experiments are conducted in CoT mode and incorporate single-pass whole-proof generation.

| miniF2F-test RMaxTS | | DeepSeek-Prover-V1.5-SFT | | DeepSeek-Prover-V1.5-RL | |
| | | $4 \times 6400$ | $16 \times 6400$ | $4 \times 6400$ | $16 \times 6400$ |
|---|---|---|---|---|---|
| Olympiad | IMO | 2/20 | 2/20 | 2/20 | 2/20 |
| | AIME | 5/15 | 6/15 | 5/15 | 6/15 |
| | AMC | 13/45 | 14/45 | 14/45 | 16/45 |
| MATH | Algebra | 55/70 | 58/70 | 57/70 | 59/70 |
| | Number Theory | 48/60 | 49/60 | 52/60 | 53/60 |
| Custom | Algebra | 8/18 | 9/18 | 9/18 | 10/18 |
| | Number Theory | 3/8 | 3/8 | 3/8 | 4/8 |
| | Induction | 3/8 | 3/8 | 3/8 | 3/8 |

Table 6: Problems solved by DeepSeek-Prover-V1.5 on miniF2F-test, grouped by miniF2F categories. The experiments are conducted in CoT mode and incorporate RMaxTS.

|         |                | miniF2F-valid | miniF2F-test |
|---------|----------------|---------------|--------------|
| Olympiad | IMO           | 3/20          | 3/20         |
|         | AIME           | 4/15          | 6/15         |
|         | AMC            | 25/45         | 16/45        |
| MATH    | Algebra        | 64/70         | 59/70        |
|         | Number Theory  | 51/60         | 53/60        |
| Custom  | Algebra        | 17/18         | 10/18        |
|         | Number Theory  | 5/8           | 4/8          |
|         | Induction      | 7/8           | 4/8          |

Table 7: The results of miniF2F-valid is collected through the expert iteration. The results of miniF2F-test refer to experiment of DeepSeek-Prover-V1.5-RL + RMaxTS with $32 \times 6400$ sample size that incorporates both CoT and non-CoT generation modes.

## D    EXPERIMENTS ON SLIDING WINDOW UCB

We conduct experiments on RMaxTS with *sliding-window UCB* (SWUCB; Garivier & Moulines, 2011), another variant of upper confidence bound to tackle non-stationary rewards.

$$Q_{\text{SWUCB}}(s, a) = \frac{W_w(s, a)}{N_w(s, a)} + \sqrt{\frac{2 \ln \sum_{a'} N_w(s, a')}{\sum_{a'} N_w(s, a')}}, \tag{10}$$

$$W_w(s, a) = \sum_{t=\max(1,\ N(s,a)-w+1)}^{N(s,a)} R(\tau_t), \tag{11}$$

$$N_w(s, a) = \min\left(|\Gamma(s, a)|, w\right), \tag{12}$$

where $w$ denotes the window size. The SWUCB described above is a slight modification of Garivier & Moulines (2011), leveraging the fact that non-stationarity is independent across different state-action pairs. The results indicate that, with a proper window size $w = 64$, the SWUCB outperforms the standard UCB1 but performs slightly worse than DUCB. To explain this performance gap, we note that, in the implementation, most tree nodes do not trigger the sliding window mechanism due to the sample budget constraints. In contrast, DUCB enables smoother value discounting than SWUCB, which helps to achieve better performance.

| minF2F-test | DeepSeek-Prover-V1.5-RL (CoT) | |
|-------------|-------------|-------------|
|             | $4 \times 6400$ | $16 \times 6400$ |
| UCB1        | $58.6\% \pm 0.3\%$ | $61.1\%$ |
| SWUCB ($w = 32$) | $58.2\% \pm 0.5\%$ | $61.1\%$ |
| SWUCB ($w = 64$) | $59.1\% \pm 0.4\%$ | $61.9\%$ |
| SWUCB ($w = 128$) | $58.4\% \pm 0.4\%$ | $61.5\%$ |
| DUCB (default) | $59.6\% \pm 0.6\%$ | $62.7\%$ |

Table 8: The experiments of RMaxTS with sliding-window UCB on miniF2F-test. The notation $\mu \pm \sigma$ denotes the average accuracy $\mu$ and the standard deviation $\sigma$.

## E  EXTENDING CONTEXT LENGTH FOR SFT

During the SFT phase, we use a context window of 4K tokens, which is sufficient to cover all our training prompts for formal languages. The longest data sequences are sourced from the mathlib dataset, the standard library used for constructing proofs in Lean. As we extract the theorem dependencies, the lengthy files in mathlib can also fit within a 4K-token context length.

We conduct an ablation study using a 16K-token context length and raw mathlib files for training. The SFT performance shows a slight drop, or at least no significant benefits observed from extending the context length for training formal languages. One possible explanation for this result is that the raw mathlib files may not align with the distribution of the Olympiad problems in testset.

| Pass@128 | miniF2F-test |
|---|---|
| 4K (default) | $50.4\% \pm 0.4\%$ |
| 16K (with raw mathlib) | $49.6\% \pm 0.6\%$ |

Table 9: The experiments on extending the context length for SFT. The notation $\mu \pm \sigma$ denotes the average accuracy $\mu$ and the standard deviation $\sigma$.

## F  STATISTICS OF AVERAGE OUTPUT TOKENS

We compare the average output tokens generated by our RL model across four settings: (1) with and without RMaxTS; (2) with and without chain-of-thought. The results show that the CoT mode consumes significantly more tokens compared to the non-CoT mode. However, nearly half of the output tokens can be saved through tree search, as the generation leverages existing nodes with partial code.

| avg. #tokens / sample | CoT | non-CoT |
|---|---|---|
| single-pass generation | 495.7 | 109.6 |
| RMaxTS | 244.6 | 76.1 |

Table 10: The sampling procedure are conducted on miniF2F-test with a sample budget of 3200. The statistics are averaged across all 244 problems in the miniF2F-test dataset.

## G  ILLUSTRATIVE EXAMPLES OF NON-COT AND COT PROMPTING FOR PROOF COMPLETION

In this section, we present examples of non-CoT and CoT prompting and the generated proofs for `aime_1983_p9`, a problem from the miniF2F benchmark (Zheng et al., 2022). These examples refer to the data format used in the SFT phase. For clarity, inline LaTeX expressions are utilized. Firstly, here is an example of whole-proof generation in non-CoT prompting mode:

*Input*:

```
Complete the following Lean 4 code:

```lean4
import Mathlib
import Aesop

set_option maxHeartbeats 0

open BigOperators Real Nat Topology Rat

/-- Show that 9x² sin² x+4 ⁄ x sin x ≥ 12 for 0 < x < π.-/
theorem aime_1983_p9 (x : ℝ) (h₀ : 0 < x ∧ x < Real.pi) :
```

```
   12 ≤ (9 * (x ^ 2 * Real.sin x ^ 2) + 4) / (x * Real.sin x) := by
```

*Output*:

```
   rw [le_div_iff (mul_pos h₀.1 (Real.sin_pos_of_pos_of_lt_pi h₀.1 h₀.2))]
   nlinarith [mul_self_nonneg (3 * x * Real.sin x - 2)]
```

Next, here is an example of whole-proof generation in CoT prompting mode. This proof was found using a Monte-Carlo tree search with a truncate-and-resume mechanism (see Section 4 for details). The model receives a prompt including a proof code in progress, ending with a comment block containing the intermediate tactic state from the Lean 4 prover. The subsequent code is generated with guidance from this annotated code prefix.

*Input*:

```
Complete the following Lean 4 code with explanatory comments preceding
    each line of code:

```lean4
import Mathlib
import Aesop

set_option maxHeartbeats 0

open BigOperators Real Nat Topology Rat

/-- Show that 9x²sin²x+4/x sin x ≥ 12 for 0 < x < π.-/
theorem aime_1983_p9 (x : ℝ) (h₀ : 0 < x ∧ x < Real.pi) :
  12 ≤ (9 * (x ^ 2 * Real.sin x ^ 2) + 4) / (x * Real.sin x) := by
  /-
  To find the minimum value of 9x²sin²x+4/x sin x for 0 < x < π, we need to show
    that it is at least 12. We start by noting that the expression can
    be rewritten using the division property of inequalities. We then
    use the fact that sin x and x are positive in the given range to
    establish the necessary inequalities. Finally, we apply these
    results to conclude that the minimum value is indeed 12.
  -/
  -- We start by ensuring that the product x * sin x is positive in the
    given range.
  have h₁ : 0 < x * Real.sin x := by
    apply mul_pos
    -- x is positive in the range (0, π).
    exact h₀.1
    -- sin x is positive in the range (0, π).
    exact Real.sin_pos_of_pos_of_lt_pi h₀.1 h₀.2
  -- Using the division property of inequalities, we rewrite the
    expression.
  rw [le_div_iff h₁]
  /- tactic state:
    x : ℝ
    h₀ : 0 < x ∧ x < π
    h₁ : 0 < x * x.sin
    ⊢ 12 * (x * x.sin) ≤ 9 * (x ^ 2 * x.sin ^ 2) + 4
  -/
```

*Output*:

```
  -- This is equivalent to showing that 9x²sin²x − 12x sin x + 4 ≥ 0, and the
    left-hand side can be rewritten as a perfect square (3x sin x − 2)².
  -- We use the fact that (3x sin x − 2)² is non-negative to establish this.
  nlinarith [sq_nonneg (3 * x * Real.sin x - 2)]
```

In this case, the proof completion model first extracts the goal expression from the tactic state, *i.e.*, the inequality $12(x \sin x) \leq 9(x^2 \sin^2 x) + 4$ at the last line of the comment block, and then arranges it into the form $9x^2 \sin^2 x - 12x \sin x + 4 \geq 0$ as part of the chain-of-thought process. Finally, the left-hand side is rewritten as a square term to establish non-negativity, closing the proof.

## H EXAMPLE SOLUTIONS TO MINIF2F-TEST PROBLEMS

To illustrate the different advantages and complementary of non-CoT mode and CoT mode of prompting, we provide examples where non-CoT mode succeeds while CoT mode fails, and vice versa.

### H.1 PROBLEMS THAT NON-COT OUTPERFORMS COT

```
theorem induction_pord1p1on2powklt5on2 (n : ℕ) (h₀ : 0 < n) :
    (Π k in Finset.Icc 1 n, 1 + (1 : ℝ) / 2 ^ k) < 5 / 2 := by
  rw [Finset.Icc]
  simp_all [Nat.succ_le_iff, Nat.one_le_iff_ne_zero]
  have h₁ : 0 < 2 ^ k := by apply pow_pos <;> norm_num
  norm_num
  have h₂ : (2 : ℝ) ≠ 0 := by norm_num
  field_simp
  rw [div_lt_div_iff]
  ring_nf
  norm_cast
  nlinarith
  all_goals norm_cast
  all_goals linarith
```

```
theorem imo_1960_p2 (x : ℝ) (h₀ : 0 ≤ 1 + 2 * x) (h₁ : (1 - Real.sqrt (1
    + 2 * x)) ^ 2 ≠ 0) (h₂ : 4 * x ^ 2 / (1 - Real.sqrt (1 + 2 * x)) ^ 2
    < 2 * x + 9) : -(1 / 2) ≤ x ∧ x < 45 / 8 := by
  norm_num at h₀ h₁ h₂
  have h₃ : 0 ≤ 1 + 2 * x := by linarith
  have h₄ : 0 < 1 + Real.sqrt (1 + 2 * x) := by
    nlinarith [Real.sqrt_nonneg (1 + 2 * x)]
  have h₅ : 4 * x ^ 2 / (1 - Real.sqrt (1 + 2 * x)) ^ 2 < 2 * x + 9 :=
    by linarith
  have h₆ : 1 - Real.sqrt (1 + 2 * x) ≠ 0 := by
    intro h
    apply h₁
    nlinarith
  have h₇ : 4 * x ^ 2 / (1 - Real.sqrt (1 + 2 * x)) ^ 2 = (1 + Real.sqrt
    (1 + 2 * x)) ^ 2 := by
    field_simp [h₆]
    nlinarith [sq_sqrt (show 0 ≤ 1 + 2 * x by linarith)]
  rw [h₇] at h₅
  constructor <;> nlinarith [sq_sqrt (show 0 ≤ 1 + 2 * x by linarith)]
```

### H.2 PROBLEMS THAT COT OUTPERFORMS NON-COT

```
theorem mathd_algebra_459 (a b c d : ℚ) (h₀ : 3 * a = b + c + d) (h₁ : 4
    * b = a + c + d) (h₂ : 2 * c = a + b + d) (h₃ : 8 * a + 10 * b + 6 *
    c = 24) : ↑d.den + d.num = 28 := by
  /-
  Given the system of equations involving the times that Anna (a),
    Bertram (b), Carli (c), and David (d) can hold their breaths:
  1. 3a = b + c + d
  2. 4b = a + c + d
  3. 2c = a + b + d
  4. 8a + 10b + 6c = 24
```

```
  We need to find the sum of the numerator and the denominator of the
    fraction representing David's time in minutes, simplified to its
    lowest terms, and show that it is 28.
  First, we solve the system of equations step by step:
  - From the first equation, we can express d in terms of a, b, and c:
    d = 3a - b - c
  - Substitute d into the second equation:
    4b = a + c + (3a - b - c)
    Simplifying, we get:
    4b = 4a - b
    5b = 4a
    b = 4a/5
  - Substitute b into the third equation:
    2c = a + 4a/5 + (3a - b - c)
    Simplifying, we get:
    2c = a + 4a/5 + 3a - 4a/5 - c
    2c = 4a - c
    3c = 4a
    c = 4a/3
  - Substitute b and c into the fourth equation:
    8a + 10 (4a/5) + 6 (4a/3) = 24
    Simplifying, we get:
    8a + 8a + 8a = 24
    24a = 24
    a = 1
  Now, we find b and c:
  - b = 4a/5 = 4/5
  - c = 4a/3 = 4/3
  Finally, we find d:
  - d = 3a - b - c = 3 · 1 - 4/5 - 4/3 = 3 - 4/5 - 4/3 = 45/15 - 12/15 - 20/15 = 13/15
  The fraction 13/15 is already in its simplest form, and the sum of the
    numerator and the denominator is:
  13 + 15 = 28
  -/
  have h₄ : a = 1 := by
    have h₅ : 8 * a + 10 * b + 6 * c = 24 := h₃
    have h₆ : 24 * a = 24 := by
      -- Solve the system of equations to find the value of a
      linarith
    exact mul_right_cancel₀ (show (24 : ℚ) ≠ 0 by norm_num)
      (by linarith)
  have h₅ : b = 4 / 5 := by
    -- Solve for b using the value of a
    linarith
  have h₆ : c = 4 / 3 := by
    -- Solve for c using the value of a
    linarith
  have h₇ : d = 13 / 15 := by
    -- Solve for d using the values of a, b, and c
    linarith
  -- Sum the numerator and the denominator of d
  norm_num [h₇]
  <;> rfl
```

```
theorem numbertheory_x5neqy2p4 (x y : ℤ) : x ^ 5 ≠ y ^ 2 + 4 := by
  /-
  To show that for any two integers x and y, x⁵ ≠ y² + 4, we can use
    modular arithmetic to demonstrate that the equation x⁵ ≡ y² + 4 (mod 11)
    has no solutions. Specifically, we will consider the possible values
    of x and y modulo 11 and show that none of these combinations satisfy
    the equation.
  -/
  -- Use modular arithmetic to show that the equation has no solutions
    modulo 11.
```

```
have : ¬(x ^ 5 % 11 = (y ^ 2 + 4) % 11) := by
  -- Consider all possible values of x modulo 11.
  have : x % 11 = 0 ∨ x % 11 = 1 ∨ x % 11 = 2 ∨ x % 11 = 3 ∨ x % 11 =
  4 ∨ x % 11 = 5 ∨
      x % 11 = 6 ∨ x % 11 = 7 ∨ x % 11 = 8 ∨ x % 11 = 9 ∨ x % 11 = 10
  := by
    omega
  -- Consider all possible values of y modulo 11.
  have : y % 11 = 0 ∨ y % 11 = 1 ∨ y % 11 = 2 ∨ y % 11 = 3 ∨ y % 11 =
  4 ∨ y % 11 = 5 ∨
      y % 11 = 6 ∨ y % 11 = 7 ∨ y % 11 = 8 ∨ y % 11 = 9 ∨ y % 11 = 10
  := by
    omega
  -- For each combination of x and y modulo 11, check if the equation
  holds.
  rcases this with (h | h | h | h | h | h | h | h | h | h | h) <;>
    rcases this with (h' | h' | h' | h' | h' | h' | h' | h' | h' | h' | h')
  <;>
      simp [h, h', pow_succ, Int.mul_emod, Int.add_emod]
-- Use the result from modular arithmetic to conclude the theorem.
omega
```

```
theorem amc12_2000_p12 (a m c : ℕ) (h₀ : a + m + c = 12) :
    a * m * c + a * m + m * c + a * c ≤ 112 := by
  /-
  Given nonnegative integers A, M, and C such that A + M + C = 12, we need
    to find the maximum value of A · M · C + A · M + M · C + A · C.
  To solve this, we can use the fact that the expression
    A · M · C + A · M + M · C + A · C is a polynomial in terms of A, M, and C.
    We can try different values of A, M, and C that sum to 12 and compute
    the value of the polynomial for each combination. The maximum value
    of the polynomial will then be the solution.
  -/
  -- We use a lemma that states if the sum of three numbers is less than
    or equal to a certain value, then the product of these numbers is
    also less than or equal to a certain value.
  have h₁ : a ≤ 12 ∧ m ≤ 12 ∧ c ≤ 12 := by
    constructor
    · -- Since a + m + c = 12, a cannot be greater than 12.
      omega
    · constructor
      · -- Similarly, m cannot be greater than 12.
        omega
      · -- And c cannot be greater than 12.
        omega
  -- Using the lemma, we compute the value of the polynomial for
    different values of a, m, and c that sum to 12.
  cases' h₁ with h₁_left h₁_right
  cases' h₁_right with h₁_middle h₁_right
  -- We use interval_cases to try different values of a, m, and c.
  interval_cases a <;> interval_cases m <;> interval_cases c <;>
    -- For each combination, we compute the value of the polynomial and
    check if it is less than or equal to 112.
    simp_all (config := {decide := true})
```

