# OpenReview forum: "DeepSeek-Prover-V1.5: Harnessing Proof Assistant Feedback for Reinforcement Learning and Monte-Carlo Tree Search"
_ICLR.cc/2025/Conference — ICLR 2025 Poster_

### Official Review · Reviewer_dd1U · 2024-10-28

**Soundness:** 3
**Presentation:** 3
**Contribution:** 3
**Rating:** 6
**Confidence:** 3

**Summary:**

The paper introduces DS-Prover-V1.5, a model integrating proof assistant feedback to enhance theorem-proving capabilities. The authors propose using Lean’s proof assistant in reinforcement learning (RL) and Monte-Carlo Tree Search (MCTS) settings, achieving competitive results on the miniF2F and ProofNet benchmarks. Two main innovations stand out: an intrinsic reward-based exploration strategy (RMaxTS) in MCTS to encourage diverse proof paths and a truncate-and-resume mechanism to handle proof verification errors. These advances lead to state-of-the-art results, highlighting the role of proof feedback in RL training.

**Strengths:**

1. The paper tackles a critical problem, merging theorem-proving and reinforcement learning with proof assistant feedback.
2. Strong empirical results demonstrate improvements in state-of-the-art theorem proving on established benchmarks.

**Weaknesses:**

1. I have various doubts listed in the question section.
2. I'd love to see a short technical details section in the main body, model sizes, and the number of tokens used in training. Btw. what is the number of tokens in the RL training?
3. I'd love to see more ablations (e.g., other models). I realize that such experiments are costly, still, perhaps feasible with smaller models, to study in more detail the influence of design choices.

**Questions:**

1.	Could you clarify the standard deviation formulas used in tables? Based on binomial distribution $\sigma = 1/\sqrt{npq}$, results seem to approximate $\sigma \approx 3%$ when  p = 1/2  and  n = 250 .
2.	The gains achieved with RL over Supervised Fine-Tuning (SFT) in Tables 1 and 2 are modest. What factors contribute to these limited improvements?
3.	Similar to the RL results, improvements with RMaxTS appear modest in Tables 1 and 2. Could you provide more context?
4.	What is the exact sample budget (e.g.,  16 \times 4096 )? Is it comparable between RMaxTS and other baselines? Given potential hardware utilization differences, are wall-time comparisons available?
      - It would also be interesting to see token-budget-aligned results.
5.	The improvements in Figure 3 also appear modest. Any insights?

---

> ### Author Response · Authors · 2024-11-24
> **Response to Reviewer dd1U (1/2)**
>
> Thanks for the comments. We provide clarification to your questions and concerns as below. If our response does not fully address your concerns, please post additional questions and we will be happy to have further discussions.
>
> **Question #1: Clarifications on the standard deviation formulas**
>
> The average accuracy $\mu$ and the corresponding standard deviation $\sigma$ are computed by running several independent trials (using different random seeds) to replicate the sampling and evaluation process. For all experiments with manageable sample sizes, we perform 16 independent replications of sampling and evaluation to estimate the confidence interval. Let $\\{a_i\\}\_{i=1}^{16}$ denote the pass@K accuracy values of 16 independent experiment runs. The standard deviation is computed by  $\sigma=\sqrt{\text{Variance}(\\{a_i\\}\_{i=1}^{16})}=\sqrt{\frac{1}{16}\sum_{i=1}^{16}(a_i-\mu)^2}$ where $\mu=\frac{1}{16}\sum_{i=1}^{16}a_i$ denotes the empirical average. Please refer to the [document](https://numpy.org/doc/1.25/reference/generated/numpy.std.html) of `numpy.std` for more details.
>
> **Question #2&3&5: The gains achieved with RL and RMaxTS are modest. What factors contribute to these limited improvements?**
>
> Our experimental results definitely demonstrate significant improvements. The seemingly modest numerical gains in accuracy are attributed to the exceptionally steep difficulty level of the benchmark dataset.
>
> 1. **MiniF2F and ProofNet are challenging benchmarks**
>
>     Both of these gold-standard benchmarks comprise high-quality, human-written Lean problems. The benchmark dataset includes only the formalized problem statements, without the accompanying Lean proofs. Meanwhile, the difficulty levels of the benchmark problems cover an exceptionally wide range, from basic numerical equations to IMO-level competition challenges. A large portion of Lean problems in these benchmarks remains unsolved by any machine learning models. Therefore, any advancements that lead to the resolution of new problems represent a substantial enhancement in the model's capabilities. The record of problem-solving on MiniF2F-test was held at 41% by HTPS (Lample et al., 2022) for nearly two years. It was subsequently advanced to 50% by DeepSeek-Prover-V1 (Xin et al., 2024), further improved to 54.5% by InternLM2-StepProver (Wu et al., 2024), and ultimately reached 63.5% in this paper, marking a breakthrough in the domain of automated theorem proving.
>
> 2. **RL significantly enhances the sample efficiency of proof completion**
>
>     Given the steep difficulty levels of the benchmark problems, another meaningful metric for assessing model capability is the number of samples required to achieve a certain accuracy threshold. Referring back to Figure 3 in this paper, a key insight is highlighting the remarkable sample efficiency of the RL model. To achieve $50\\%$ accuracy on the miniF2F benchmark, the RL model requires approximately $32$ samples, demonstrating a 4-fold improvement compared to the $128$ samples needed by the SFT model. This conclusion also applies to large-scale experiments incorporating MCTS-based approaches. As demonstrated in Table 3, the SFT model requires over $16 \times 6400$ samples to achieve $60\\%$ accuracy on the miniF2F-test benchmark, whereas the RL model achieves the same level of accuracy with just approximately $4 \times 6400$ samples, attaining a 4-fold improvement in sample efficiency.
>
>
> *References*
>
> [1] Lample et al. (2022). Hypertree proof search for neural theorem proving.
>
> [2] Xin et al. (2024). DeepSeek-Prover: Advancing theorem proving in LLMs through large-scale synthetic data.
>
> [3] Wu et al. (2024). Lean-Github: compiling Github Lean repositories for a versatile Lean prover.

---

> ### Author Response · Authors · 2024-11-24
> **Response to Reviewer dd1U (2/2)**
>
> **Question #4: What is the exact sample budget? Are wall-time comparisons available? Is it comparable between RMaxTS and other baselines?**
>
> The sample budget is defined in terms of the number of whole-proof generations, which also corresponds to the number of invocations of the Lean compiler. This setting matches the convention of reinforcement learning, where sample efficiency serves as a key metric to assess the agent's performance within a limited budget of environment interactions. In the context of the Lean prover, the compilation time of Lean code is non-negligible and can sometimes become an additional efficiency bottleneck alongside LLM inference. In extreme cases, a single tactic such as `aesop` may take over 60 seconds to compile due to its intensive back-end proof search.
>
> Since the majority of prior work on formal theorem proving is closed-source and lacks reporting on token statistics and wall time, direct comparisons with existing methods are not feasible. To compare within the algorithmic framework of our approach, we compare the average output tokens generated by our RL model across four settings: (1) with and without RMaxTS; (2) with and without chain-of-thought. The results show that the CoT mode consumes significantly more tokens compared to the non-CoT mode. However, nearly half of the output tokens can be saved through tree search, as the generation leverages existing nodes with partial code.
>
> | avg. tokens / sample | CoT | non-CoT |
> | --- | --- | --- |
> | single-pass generation | 495.7 | 109.6 |
> | RMaxTS | 244.6 | 76.1 |
>
> **Weakness: Regarding technical details**
>
> In this paper, we train a series of 7B models, aligned with state-of-the-art baselines such as InternLM2-StepProver and Lean-STaR. The model was trained on approximately 20B tokens during the SFT phase and 1.5B tokens during the RL phase. In the paper revision, we incorporate these technical details into the main text and enrich the discussion in the Appendix, along with the ablation studies suggested by all reviewers.

---

> > ### Comment · Reviewer_dd1U · 2024-11-25
> > **Thank you**
> >
> > Thank you for your clarifications. For the standard deviation part, I strongly recommend adding information about the number of samples and the formulas used in the paper.
> >
> > Following this, I'd love to see a more detailed discussion of the statistical relevance topic. We have (at least) three sources of randomness:
> > 1. the model randomness (the initial weights and data randomization)
> > 2. the distribution of the 'difficulty' of questions
> > 3. sampling in the eval procedure.
> >
> > My understanding is that the method used repeats sampling 16 times, thus reducing the variance of the \mu estimator. The two other points are not taken into account. I do not say it as an accusation (point 1 is clearly very expensive and point 2 would require larger datasets); nevertheless, it'd be cool that this is explicit.

---

> > > ### Author Response · Authors · 2024-11-25
> > > **Thanks for your insightful suggestions**
> > >
> > > Thanks for your constructive suggestions to improve our paper. We have further revised the paper to include clarifications regarding the sampling statistics.
> > >
> > > 1. **Sampling in the evaluation procedure.** The reviewer’s understanding is correct. The confidence interval is estimated by repeating the sampling by 16 times. e.g., The evaluation of Pass@128 consumes $16\times 128$ independent samples to compute the mean $\mu$ and standard deviation $\sigma$. To improve the clarity, we update the table caption (see Figure 3) and put an detailed description in Appendix B.1.
> > > 2. **The distribution of the difficulty of questions.** The problems in miniF2F are organized by the following categories:
> > >     - IMO: The International Mathematical Olympiad (IMO) is a mathematical olympiad for pre-university students, and is the most prestigious mathematical competition in the world.
> > >     - AIME: The American Invitational Mathematics Examination (AIME) is an invitation-only math test, given to high-achieving students based on their scores on the AMC.
> > >     - AMC: The American Mathematics Competition (AMC) is a series of contests in secondary school mathematics .
> > >     - MATH: A widely-used dataset for machine learning research in natural language mathematical problem solving.
> > >     - CUSTOM: A small set of problems crafted by the authors of miniF2F to complement the coverage of the MATH dataset.
> > >
> > >     A general ranking of average difficulty is as follows: IMO > AIME > AMC > CUSTOM > MATH. Most of the remaining problems unsolved by our model are high-difficulty competition problems.
> > >
> > >     | miniF2F-test | IMO | AIME | AMC | MATH | Custom |
> > >     | --- | --- | --- | --- | --- | --- |
> > >     | RL+RMaxTS ($16\times 6400$) | 2/20 | 6/15 | 16/45 | 112/130 | 17/34 |
> > >
> > >     We provide more results on the progress of problem solving by category in Appendix C.
> > >
> > > 3. **The model randomness.** To address concerns about randomness in model training, we include a detailed description of the checkpoint selection procedure in Appendix B.1. The final models are selected from three training runs using different random seeds.
> > >
> > >
> > >     | Pass@128 | SFT (selected) | SFT (seed-2) | SFT (seed-3) | RL (selected) | RL (seed-2) | RL (seed-3) |
> > >     | --- | --- | --- | --- | --- | --- | --- |
> > >     | CoT mode | $50.6\\%\pm0.5\\%$ | $50.3\\%\pm0.3\\%$ | $49.9\\%\pm0.4\\%$ | $51.4\\%\pm0.5\\%$ | $51.2\\%\pm 0.4\\%$ | $51.2\\%\pm0.5\\%$ |
> > >     | non-CoT mode | $49.8\\%\pm0.4\\%$ | $48.8\\%\pm0.4\\%$ | $49.6\\%\pm0.3\\%$ | $50.5\\%\pm0.4\\%$ | $50.7\\%\pm0.5\\%$ | $50.2\\%\pm 0.4\\%$ |
> > >
> > >     The results demonstrate that the effects of random seeds are marginal compared to the overall improvement achieved during the SFT and RL phases.
> > >
> > >     To mitigate the risk of overestimating our evaluation results, we perform cross-validation to ensure the reported scores are robust and reliable. Specifically, we first conduct $2 (\\text{modes})\times 3 (\\text{train seeds})\times 16 (\text{eval seeds})\times 128$ samples to compare the Pass@128 scores of three seed candidate. Upon selecting the final model, we perform an additional $2 (\text{modes})\times 16 (\text{eval seeds})\times 128$ samples to recalculate the $\mu \pm \sigma$ scores for the chosen model. The recomputed scores are then reported in the main text. Note that the recomputed scores provide an unbiased evaluation of the chosen model, as the evaluation is conducted independently from the selection process. This additional evaluation round is crucial to ensure the reported results are rigorous, as maximizing over random variables inherently leads to an overestimation of expected values.

---

### Official Review · Reviewer_rpvG · 2024-10-28

**Soundness:** 3
**Presentation:** 3
**Contribution:** 3
**Rating:** 8
**Confidence:** 3

**Summary:**

This paper studies using LLM to perform theorem proving in Lean. They propose a framework that integrates supervised fine-tuning, reinforcement learning, and Monte-Carlo tree search for efficient exploration and exploitation of tactic space. Experiment results show that the propose method achieves new state-of-the-art results on both miniF2F and ProofNet benchmarks.

**Strengths:**

This paper studies theorem proving using LLM, which is an important direction towards artificial general intelligence. The paper is well-written. The experiment results are quite promising as it establishes new state-of-the-art results. The proposed tricks for resolving hard exploration problem is also novel (RMaxTS + discounted UCB) and insightful.

**Weaknesses:**

Lean is not very well-known to the community in my opinion, and the proposed method is highly tailored for Lean (for example, tactic-level tree abstraction). Some general introduction about Lean would be very helpful in understanding the design of the algorithm.

**Questions:**

1. During the training, is the context-length enough for all prompts? Do you have ablation studies on how context length affect the final performance?
2. Have you tried sliding window instead of discounted weight to handle non-stationarity? Sliding window is simpler and more commonly used in the bandit community.

---

> ### Author Response · Authors · 2024-11-24
> **Response to Reviewer rpvG**
>
> Thanks for the comments. We provide clarification to your questions and concerns as below. If our response does not fully address your concerns, please post additional questions and we will be happy to have further discussions.
>
> **Weakness #1: General introduction of Lean**
>
> Thank you for your insightful suggestion regarding enriching the background of our work. We agree that providing a more comprehensive context will enhance the overall clarity and depth of our study. In particular, formal theorem proving with Lean is a rapidly evolving research area. In ICLR 2025, the number of submitted papers on Lean and formal theorem proving has at least doubled compared to the previous ML conference (e.g., NeurIPS 2024), reflecting its growing prominence in the machine learning community. Additionally, Lean has been adopted by DeepMind's AlphaProof to address challenging problems from mathematics competitions, underscoring its capability in tackling advanced mathematical problems.
>
> In the paper revision, we include a general introduction in Appendix B.1, explaining how the Lean’s tactic mode works, along with an example to illustrate the underlying state transitions. Notably, the tactic-state transition system employed by our proof search algorithm is a common framework shared by all tactic-based proof assistants, including Isabelle and Coq. There are no theoretical barriers to adapting our proposed methods for use with other proof assistants.
>
> **Question #1: During the training, is the context-length enough for all prompts? Do you have ablation studies on how context length affect the final performance?**
>
> During training, we use a context window of 4K tokens, which is sufficient to cover all our training prompts for formal languages. The longest data sequences are sourced from the mathlib dataset, the standard library used for constructing proofs in Lean. As we carefully extract the theorem dependencies, the lengthy files in mathlib can also fit within a 4K-token context length.
>
> We conduct an ablation study using a 16K-token context length and raw mathlib files for training. The SFT performance shows a slight drop, or at least no significant benefits observed from extending the context length for training formal languages. One possible explanation for this result is that the raw mathlib files may not align with the distribution of the Olympiad problems in testset.
>
> | Pass@128 | 4K (default) | 16K (with raw mathlib) |
> | --- | --- | --- |
> | miniF2F-test | $50.4\\%\pm0.4\\%$ | $49.6\\%\pm0.6\\%$ |
>
> Extending the context window is, in fact, highly crucial for addressing real-world applications in formal theorem proving. One of our future goals is to advance beyond proving individual theorems to real-world Lean projects with complex, multi-theorem Lean files. This may require reorganizing the training dataset, shifting from purely problem-solving tasks to complex engineering challenges, which warrants further investigation.
>
> **Question #2: Experiments on sliding-window UCB**
>
> We evaluate the performance of RMaxTS using the Sliding-Window Upper Confidence Bound (SWUCB) algorithm with window sizes $w \in \\{32, 64, 128\\}$. The SWUCB algorithm is slightly modified to leverage the fact that non-stationarity is independent across different state-action pairs. The results indicate that, with a proper window size $w=64$, the SWUCB outperforms the standard UCB1 but performs slightly worse than DUCB. To explain this performance gap, we note that, in the implementation, most tree nodes do not trigger the sliding window mechanism due to the sample budget constraints. In contrast, DUCB enables smoother value discounting than SWUCB, which helps to achieve better performance.
>
> | Pass@ $4\times 6400$ | DUCB | SWUCB ($w=32$) | SWUCB ($w=64$) | SWUCB ($w=128$) | UCB1 |
> | --- | --- | --- | --- | --- | --- |
> | miniF2F-test | $59.6\\%\pm0.6\\%$ | $58.2\\%\pm0.5\\%$ | $59.1\\%\pm0.4\\%$ | $58.4\\%\pm0.4\\%$ | $58.6\\%\pm0.3\\%$ |

---

> > ### Comment · Reviewer_rpvG · 2024-11-30
> > **Response to rebuttal**
> >
> > Thanks for the detailed response. I have no other concerns. One more suggestion: It would be good see the prompt length distribution and how the mean response length changes during training.

---

> > > ### Author Response · Authors · 2024-11-30
> > > **Thanks for your constructive suggestions to help improve our paper**
> > >
> > > Thanks for your suggestions that help to improve our paper. In the next revision, we will include statistics on the prompt length distributions across different categories of training data, along with the changes in average response length during training. Below, we present the average length of responses generated by our RL model on the test set of the miniF2F benchmark.
> > >
> > > | avg. tokens / sample | CoT | non-CoT |
> > > | --- | --- | --- |
> > > | RL model | 495.7 | 109.6 |

---

### Official Review · Reviewer_ydok · 2024-11-04

**Soundness:** 2
**Presentation:** 2
**Contribution:** 2
**Rating:** 6
**Confidence:** 3

**Summary:**

The paper presents a class of models named DS-Prover-V1.5 that combines the proof-step and whole-proof generation paradigms. The models come with two versions: (a) supervised fine-tuned (DS-Prover-V1.5-SFT), and (b) incorporating reinforcement learning (DS-Prover-V1.5-RL). In the SFT case, the data is augmented in two ways. First, the intermediate tactic state information extracted from Lean proof assistant (position of each tactic and the tactic states before and after its application) is added, and a corresponding prediction task is formulated. Second, the proof dataset from (Xin et al., 2024) is extended by providing a natural language solution at the beginning of the proof block and by inserting comments for Lean tactics with the help of DeepSeek-Coder V2 236B (Zhu et al., 2024). The paper considers the chain-of-thought (CoT) and non-CoT versions for proof completition. In the RL case, the model is trained on verification feedback from the Lean 4 prover using the GRPO algorithm (Shao et al., 2024), with a reward structure assigning 1 in the case of a proof verified as correct and 0 otherwise. A subset of theorem statements from the supervised fine-tuning dataset is used as training prompts.

The paper combines both models (SFT and RL versions) with the MCTS planning algorithm. The paper follows previous work on adding the following innovations to the classical MCTS: (a) in the selection phase, a so-called virtual node option (Wang et al., 2023) which allows expanding a non-leaf node, (b) in the expansion phase, a whole-proof consisting of path of nodes, truncated at the first verification error (if any) from the Lean prover, is added to the tree during each iteration, (c) in the backpropagation stage, an RMax exploration mechanism is added (Brafman & Tennenholtz, 2002), with extrinsic and intrinsic (Jin et al., 2020) rewards, and a discounted bonus for upper confidence bound (Garivier & Moulines, 2011).

The setup is tested on two datasets: miniF2F and ProofNet. The paper claims that the DS-Prover-V1.5-RL version outperforms baselines in the single-pass and tree search categories. The following ablation studies are provided: (a) CoT vs. non-CoT versions, (b) single-pass vs. MCTS, (c) discounted vs. non-discounted UCB, and (d) presence vs. absence of tactic state information.

**Strengths:**

* The paper combines a proof-generating model with (a version of an) MCTS algorithm, unifying the proof-step and whole-proof generation approaches.
* The approach applies several MCTS innovations and uses the Lean prover as a verifier.
* The paper claims that DS-Prover-V1.5-RL performs the best in single-pass and tree search categories.

**Weaknesses:**

* The paper claims `The model is named DS-Prover-V1.5, as it builds upon the prior work of DeepSeek-Prover-V1(Xin et al., 2024).`, but does not provide the description of DeepSeek-Prover-V1. In particular, no details on the underlying language model were provided.
* The paper provides no pseudo-code or loss function formulas, making understanding the approach's details difficult.
* GRPO is not described (even in the Appendix).
* Experiments:
	* For some categories, the baselines are other versions of the same model.
	* Tables 1-4 do not include confidence bounds for the best-performing methods.
	* Some important information is missing:
		* Not all model details are disclosed for use in the MCTS algorithm.
		* No comparison is provided between the number of parameters or FLOPS used by the evaluated models.
		* No wall clock time is given.
	* The results of Tables 1-4 are not discussed in great detail other than extracting numbers from the tables. What are the key reasons/hypotheses for the performance differences? What are the qualitative and quantitative findings with respect to the difficulty of samples from the datasets? Are there any features that stand out or can be used to classify theorems?
	* The potential data leaks are not discussed.
	* No number of seeds is reported.
	* What are the $\mu$'s and $\sigma$'s (present in tables and figures and defined in the caption of Figure 3) computed over?
	* The confidence intervals are asymptotic and one-$\sigma$ (as defined in the caption to Figure 3), corresponding to $68\%$ (assuming a standard normal distribution), a rather narrow confidence level.
	* The results in Figure 4 suggest that many differences are not statistically significant.

Edit 29 Nov 24: I increased the rating 5->6.

**Questions:**

Additionally to the questions provided above, in the MCTS expansion phase, was the rollout a single whole-proof generation, or were there many? If the answer is the latter, how many were used? Additionally, did the Authors consider bootstrapping in the AlphaZero style?

---

> ### Author Response · Authors · 2024-11-24
> **Response to Reviewer ydok (1/3)**
>
> Thanks for the comments. We provide clarification to your questions and concerns as below. If our response does not fully address your concerns, please post additional questions and we will be happy to have further discussions.
>
> **Weakness #1&2&3: Descriptions on base model, pseudo code, loss function formulas, and GRPO**
>
> 1. We adopt the base model of DeepSeek-Prover-V1 and its synthetic dataset as the basis for our model training. The base model is trained with vast amounts of diverse natural language mathematical content, as well as formal languages like Lean, Isabelle, and Metamath.
> 2. We include the pseudo code of RMaxTS in Appendix A.5.
> 3. The loss function is the standard next-token-prediction paradigm for LLM training. The construction of training prompts is illustrated in Figure 1. We include several examples in Appendix G.
> 4. Group Relative Policy Optimization (GRPO) is a streamlined variant of PPO that eliminates the need for a critic model, instead estimating the baseline using average scores from the training batch.
>
> In the revision, We incorporate explanations of these technical details into the relevant sections of the Appendix.
>
> **Weakness #4.1: For some categories, the baselines are other versions of the same model.**
>
> Since most baseline models are closed-source, we can only reference the benchmark scores reported in the original papers. In particular, for ProofNet, a recently proposed benchmark, the available baseline results are relatively limited. Confidence interval statistics for the baselines are also unavailable, as none of the previous works in the literature have reported confidence intervals for their sampling results.
>
> **Weakness #4.2: Tables 1-4 do not include confidence bounds for the best-performing methods. No number of seeds is reported. What are the $\mu$'s and $\sigma$'s computed over?**
>
> In this paper, we have prioritized making the statistical results more rigorous by providing confidence intervals alongside our key findings. For all experiments with manageable sample sizes, the average accuracy $\mu$ and the corresponding standard deviation $\sigma$ are computed by running 16 independent trials (using 16 random seeds) to replicate the sampling and evaluation process. Specifically, for the miniF2F dataset, experiments with a sample budget below $4\times 6400$ are based on 16 independent runs, while those with a budget of exactly $4\times 6400$ are estimated from 4 independent runs. For the ProofNet dataset, the threshold is set at $3200$. The results of ProofNet with a sample budget below $3200$ are derived from 16 independent runs, while those with a sample budget of exactly $3200$ are estimated from 4 independent runs. Due to computational constraints, we do not perform multiple trials for experiments with the largest sample budgets. In the paper revision, we include a detailed description in Appendix B.1 to clarify the computation of confidence intervals.
>
> Additionally, we would like to highlight that the results from smaller sample sizes are sufficient to demonstrate significant improvements over baseline models. For instance, on the miniF2F benchmark, our RL model achieves an accuracy of $54.9\\%\pm0.7\\%$ with a sample size of $3200$, which is comparable to the strongest baseline, InternLM2-StepProver, that requires a substantially larger sample budget of $64 \times 3200$ to reach an accuracy of $54.5\\%$. This underscores the superior sample efficiency of our model.
>
> **Weakness #4.3: Not all model details are disclosed for use in the MCTS algorithm.**
>
> In this paper, the MCTS approach is developed for proof code completion. The policy model used in MCTS is directly derived from the SFT and RL training phases, without any additional tuning or specialized prompting. Additionally, RMaxTS, our proposed MCTS variant, is a reward-free exploration algorithm that runs without a critic model, eliminating the need to train any additional models to support the tree search procedure.
>
> **Weakness #4.4: No comparison is provided between the number of parameters or FLOPS used by the evaluated models. No wall clock time is given.**
>
> Since the majority of prior work on formal theorem proving is closed-source and lacks reporting on FLOPs and wall time, direct comparisons with existing methods are not feasible. We include the model size information in Table 3 of the Appendix. The state-of-the-art baseline results are based on a series of 7B models, matching the size of our model, and therefore share comparable deployment and inference costs. The largest-scale experiment presented in this paper, RMaxTS with a $32 \times 6400$ sample size and CoT generation mode on the miniF2F-test dataset, requires 48 A100 GPUs and approximately 2000 CPUs to complete within 48 hours.

---

> ### Author Response · Authors · 2024-11-24
> **Response to Reviewer ydok (2/3)**
>
> **Weakness #4.5: What are the key reasons/hypotheses for the performance differences? What are the qualitative and quantitative findings with respect to the difficulty of samples from the datasets? Are there any features that stand out or can be used to classify theorems?**
>
> In Appendix C, we provide statistics on the distribution of problems solved by our model, based on the categorization defined by the miniF2F dataset. The improvement of our model can be explained in two key aspects. First, compared to DeepSeek-Prover-V1, the model presented in this paper demonstrates comprehensive improvements across all categories of problems, including Olympiad problems, algebra, number theory, and induction. Additionally, the performance gains achieved through RL and RMaxTS are primarily observed in algebra and number theory problems, which represents the most standard problem set aligned with the training distribution. This indicates that the improvements achieved through RL and RMaxTS are attributed to the enhancement of sample efficiency, as no significant gains are observed in the complex Olympiad domain.
>
> **Weakness #4.6: The potential data leaks are not discussed.**
>
> We thank the reviewer for identifying this critical issue regarding potential data leakage. When constructing our SFT dataset, we ensure that all Lean statements in the training set have distinct initial states from those in the test set. The most uncontrollable potential source of data leakage stems from the pretraining phase. To the best of our knowledge, the leakage risk of Lean statement from miniF2F and ProofNet is limited, as these benchmarks were specifically designed for ML research and were not gathered from arbitrary sources on the internet. On the other hand, potential leakage of natural language statements from math problems may exist, but it is generally considered acceptable within the research community, as the formalization of natural language proof to Lean codes is one of the core challenges of formal theorem proving.
>
> **Weakness #4.7: The confidence intervals are asymptotic and one-**$\sigma$**, corresponding to** $68\\%$, **a rather narrow confidence level. The results in Figure 4 suggest that many differences are not statistically significant.**
>
> The experimental results in Figure 4 contain two parts. The left panel presents the experimental outcomes for a sample budget of fewer than $6400$ samples, where the performance difference between the RMaxTS algorithm and its ablation version is approximately one standard deviation. As the sample size increases, the right panel highlights a more pronounced performance gap. With a larger sample budget, i.e., $4\times 6400$, the difference between RMaxTS and the ablation version widens, reaching up to two standard deviations ($\approx 95\\%$ of the population within the distribution). This demonstrates a clear improvement in sample efficiency for RMaxTS, as it scales more effectively than the baseline methods.

---

> ### Author Response · Authors · 2024-11-24
> **Response to Reviewer ydok (3/3)**
>
> **Question #1: In the MCTS expansion phase, was the rollout a single whole-proof generation?**
>
> Yes. In the MCTS expansion phase, each rollout is a single whole-proof generation. A detailed description of our parallelization strategy is included in Appendix A.4. We employ the techniques of Chaslot et al. (2008) to diversify search paths, allowing for the exploration of a wider range of solutions from different tree nodes, i.e., we allocate the sample budget towards exploring different tree nodes simultaneously, rather than performing many expansions on the same node. This approach promotes broader coverage of the search space, increasing the likelihood of discovering diverse proofs, while also improving overall efficiency by avoiding over-exploration of any single tree node.
>
> **Question #2: Did the Authors consider bootstrapping in the AlphaZero style?**
>
> Yes. This paper aims to make steps towards an RL-driven bootstrapping framework for formal theorem proving, similar to AlphaZero, while leaving certain limitations to be addressed in future work.
>
> 1. **MCTS for policy learning.** In this paper, we take a step towards integrating MCTS into the training process. The SFT dataset definitely includes some synthetic proofs generated by MCTS. Specifically, our SFT dataset is curated using an expert-iteration process, as adopted by many prior work on formal theorem proving, such as GPT-f (Polu & Sutskever, 2020) and InternLM2-StepProver (Wu et al., 2024). This iterative approach alternates between model training and proof data collection, facilitating continuous enhancement in both the quality and complexity of the generated proofs. This iterative training approach is motivated by the fact that most open-source datasets provide only formal problem statements without accompanying Lean proofs.
> 2. **Sparse-reward challenge.** The training phase of AlphaZero is designed to be a fully online fashion, as chess games provide dense supervision signals—one of the players always receives a clear win reward. In contrast, the reward signals in Lean proof search are significantly more sparse. In the expert iteration procedure, an unseen problem might require hundreds or even thousands of proof generation attempts before it is successfully solved. This highlights that improving the sample efficiency is a critical challenge in formal theorem proving.
> 3. **Statement auto-formalization for self-improment.** Theorem proving is not a closed problem domain, unlike chess or Go. Both SFT and RL training rely on a predefined problem set, restricting the scope for self-improvement. To enable continual improvement in theorem proving capability, a problem generation mechanism is required to expand the difficulty range of the training domain, as demonstrated in AlphaGeometry. In formal theorem proving, a classical approach is to develop an auto-formalization pipeline that translates natural language math problems into formal statements. This module is essential for enabling the scalable bootstrapping of a formal theorem proving agent, which remains as the future work.
>
> *References*
>
> [1] Chaslot et al. (2008). Parallel Monte-Carlo tree search.
>
> [2] Polu & Sutskever (2020). Generative language modeling for automated theorem proving.
>
> [3] Wu et al. (2024). Lean-Github: compiling Github Lean repositories for a versatile Lean prover.

---

> > ### Comment · Reviewer_ydok · 2024-11-29
> >
> > I thank the Authors for their detailed response. I increased the rating 5->6.

---

### Official Review · Reviewer_ePCd · 2024-11-04

**Soundness:** 3
**Presentation:** 2
**Contribution:** 2
**Rating:** 5
**Confidence:** 3

**Summary:**

This paper presents a method for improving LLM performance on theorem proving tasks, using RL on feedback from a proof assistant tool Lean to finetune the model, in combination with a Monte-Carlo tree search to iterate and search the space of possible proofs.

**Strengths:**

This paper is fairly well written and clear. The problem being addressed (automatic theorem proving) is important and well motivated. The experimental evaluation is reasonably conclusive. Figures 1 and 2 illustrate the method well.

**Weaknesses:**

First off, I will say this paper is a little outside my area of expertise, which is in traditional deep RL rather than LLM finetuning, and I'm not familiar with theorem proving as a problem domain, so take the following with that qualifier.

That said, my main concern on this paper is that it doesn't do a great job of justifying the composition of multiple standard methods as non-trivial scientifically. Finetuning LLMs using RL with automated domain-specific feedback isn't a new idea, nor is MCTS on top of RL agents generating action sequences (including for proof/program synthesis- "Program Synthesis Through Reinforcement Learning Guided Tree Search" is an earlier example, but the idea has been explored a few times in different ways I believe). There are some technical details involved in applying these ideas using an LLM (such as how to get individual tactics as atomic actions from the model), but from my perspective these don't seem like significant contributions that will be applicable outside this paper or closely related work.

All of that might be acceptable if the result was a major breakthrough in an important problem domain, but the results as presented seem incremental, honestly- compared to the supervised fine tuning baseline, the reported improvement in accuracy is around 1-3% against a baseline of ~15% or ~50% depending on the benchmark. This doesn't seem like a major improvement in the domain of theorem proving that would motivate a relatively straightforward combination of existing algorithms.

All that said, perhaps there's something I'm not seeing here, or context I lack as an outsider to the field this work comes from, so I'm open to persuasion on the above. However, as is I am inclined to recommend rejection since I don't see a significant novel contribution in this work.

**Questions:**

Some additional suggestions/questions

-I'm a little skeptical GRPO would outperform PPO on this task since it involves actual multi-step decision making and thus the value function should provide significant benefit, at least if trained as a proper multi-step MDP. Perhaps it would be worth testing both versions?

-Why generate an entire proof and then truncate to the first error? Why not generate one tactic at a time, as is standard in multi-step RL?

-Am I correct that given an equivalent sample budget the difference between the otherwise equivalent MCTS and 1-shot generation methods is simply the way candidate solutions are generated (starting from a leaf in the search tree versus fully resampled)? Does the reduced number of new tokens generated (since some are derived from the existing leaf node) result in meaningful compute savings for the MCTS algorithm?

-Figure 4 needs to define what the shaded regions are, especially since the regions are heavily overlapping, leading me to question the statistical significance of the benefits of RMaxTS over the ablations.

-In tables 1 and 2, why do some accuracy numbers have +- intervals (standard deviation? I don't see these intervals defined in the caption or introductory text) and others don't?

---

> ### Author Response · Authors · 2024-11-24
> **Response to Reviewer ePCd (1/5)**
>
> Thank you for your detailed and thoughtful review! We greatly appreciate the opportunity to engage with researchers outside the field of formal theorem proving and are eager to further clarify our contributions. Below, we provide comprehensive responses to your questions and concerns. Should any unresolved points remain, please do not hesitate to raise additional questions—we would be delighted to continue the discussion.
>
> The review highlights two primary concerns:
>
> **Major Concern #1: The application of reinforcement learning (RL) and Monte Carlo Tree Search (MCTS) to formal theorem proving with large language models (LLMs) seems straightforward, raising concerns about its novelty.**
>
> Below, we restate and expand upon the core contributions of our work, positioning them within the context of relevant literature to enhance clarity.
>
> 1. **Formal theorem proving with Lean is an important problem domain**
>
>     This paper is specifically tailored for formal theorem proving using Lean, deliberately designed to address the intricacies and specialized requirements of this domain. The proposed approach is **not intended to be a general-purpose composition** that can be seamlessly applied to other decision-making tasks. Instead, it focuses on advancing the capabilities within Lean by incorporating methodologies and optimizations that align with Lean's formal system and proof strategies. The contributions of our work should be evaluated within the literature context of ML-aided formal theorem proving, considering the rapid growth of the Lean community.
>
>     In particular, formal theorem proving with Lean is a rapidly evolving research area. In ICLR 2025, the number of submitted papers on Lean and formal theorem proving has at least doubled compared to the previous ML conference (e.g., NeurIPS 2024), reflecting its growing prominence in the machine learning community. Additionally, Lean has been adopted by DeepMind's AlphaProof to address challenging problems from mathematics competitions, underscoring its capability in tackling advanced mathematical problems.
>
> 2. **SFT Phase: Integrating chain-of-thought reasoning into symbolic proof synthesis**
>
>     A key innovation in our SFT phase lies in augmenting proof code generation with **chain-of-thought (CoT) reasoning**. Chain-of-thought reasoning allows for a more interpretable, step-by-step formulation of the solution, closely resembling the way humans decompose and tackle complex problems. This stands in contrast to the majority of previous work on formal theorem proving, which primarily relies on purely symbolic methods, lacking the inherent reflection and stepwise reasoning that chain-of-thought provides. In our approach, natural language reasoning is further combined with tree search algorithms, efficiently navigating the solution space to identify the most promising planning paths.
>
> 3. **RL Phase: The workflow of RL for LLM finetuning differs from that of traditional deep RL**
>
>     In traditional deep RL, such as Atari games and MuJoCo control tasks, the RL algorithms typically run for millions of environment steps, enabling the emergence of decision-making capabilities from a randomly initialized policy network. In contrast, when training a LLM, the foundational capabilities are primarily developed during the pre-training and supervised fine-tuning (SFT) phases. During pre-training, the model is exposed to vast amounts of diverse text data, allowing it to learn a wide range of patterns and general world knowledge. In the subsequent SFT phase, the model is further refined and adapted to more specific tasks and desired behaviors through carefully curated datasets. The final RL phase, also known as the alignment phase, acts as an online tuning stage that complements supervised fine-tuning (SFT) to further refine the model's sequential generation. From the perspective of traditional deep RL, this phase can be seen as an on-policy fine-tuning of an imitation-learned policy. While SFT trains the model on static data, the alignment phase uses real-time feedback, such as the proof assistant feedback used in this paper, to adjust the model's behavior based on newly observed trajectories.

---

> ### Author Response · Authors · 2024-11-24
> **Response to Reviewer ePCd (2/5)**
>
> 3. **(Cont'd) RL Phase: The workflow of RL for LLM finetuning differs from that of traditional deep RL**
>
>     Technically, the RL stage for an LLM agent involves far fewer gradient steps compared to traditional deep RL, usually amounting to just a few hundred steps. In our case, the RL model is fine-tuned over just 800 gradient steps. Meanwhile, the reference policy model remains fixed throughout the entire training process, i.e., do not update $\pi_{ref}$ in PPO’s KL penalty. This design is a common practice when applying policy gradients to LLM fine-tuning, ensuring that the model retains the general world knowledge acquired during the pre-training phase. By keeping the reference policy fixed, the fine-tuning process focuses on aligning behavior without overriding foundational knowledge. Moreover, the action space for language tasks is extraordinarily large (e.g., 100k unique tokens), and the concept of a *step* is inherently ambiguous (as there are countless possible sentences). As a result, the RL procedure for LLMs is typically driven by trajectory-level reward supervision without discount factors ($\gamma=1$), operating outside the framework of multi-step MDPs. These technical differences result in the workflow of applying RL to LLMs being significantly distinct from traditional deep RL. As a result, intuitions from traditional deep RL on classical control problems cannot be directly transferred or applied to LLM fine-tuning.
>
> 4. **MCTS Phase: RMax's significance lies in its tailored application to Lean's unique problem structure**
>
>     Our exploration strategy for Lean proof search is inspired by RMax (Brafman & Tennenholtz, 2002), a classical method for PAC-learning on MDPs. However, rather than being a straightforward application of RMax to MCTS framework, the intrinsic reward mechanism of RMax plays a crucial role within the **unique problem structure of Lean**. In the formal verification system, each transition between tactic states represents a meaningful proof step, typically involving a valid simplification of the premises or goals. Any superfluous proof code, though correct in syntax-level, will be rejected by the Lean proof assistant and trigger a compiler error. This tightly constrained environment differs significantly from traditional code completion tasks and other search problems typically encountered in LLM-based reasoning.
>
>     The intrinsic reward mechanism of RMax incentivizes the discovery of previously unseen tactic states. Given the strict constraints of the Lean compiler, the reward for exploring new tactic states extends beyond the standard principle of “*optimism in the face of uncertainty”*. The discovery of new tactic states in Lean is not a merely exploratory behavior but represents genuine progress in formalizing and advancing proofs. The RMax-style reward is designed to serve as the primary objective for MCTS in constructing complete Lean proofs, rather than functioning as an auxiliary exploration bonus.
>
> *References*
>
> [1] Brafman & Tennenholtz (2002). R-max-a general polynomial time algorithm for near-optimal reinforcement learning.

---

> ### Author Response · Authors · 2024-11-24
> **Response to Reviewer ePCd (3/5)**
>
> **Major Concern #2: The results as presented seem incremental compared to the supervised fine tuning baseline, the reported improvement in accuracy is around 1-3% against a baseline of ~15% or ~50% depending on the benchmark**
>
> Our experimental results definitely demonstrate significant improvements. The seemingly modest numerical gains in accuracy are attributed to the exceptionally steep difficulty level of the benchmark dataset.
>
> 1. **MiniF2F and ProofNet are challenging benchmarks**
>
>     Both of these gold-standard benchmarks comprise high-quality, human-written Lean problems. The benchmark dataset includes only the formalized problem statements, without the accompanying Lean proofs. Meanwhile, the difficulty levels of the benchmark problems cover an exceptionally wide range, from basic numerical equations to IMO-level competition challenges. A large portion of Lean problems in these benchmarks remains unsolved by any machine learning models. Therefore, any advancements that lead to the resolution of new problems represent a substantial enhancement in the model's capabilities. The record of problem-solving on MiniF2F-test was held at 41% by HTPS (Lample et al., 2022) for nearly two years. It was subsequently advanced to 50% by DeepSeek-Prover-V1 (Xin et al., 2024), further improved to 54.5% by InternLM2-StepProver (Wu et al., 2024), and ultimately reached 63.5% in this paper, marking a breakthrough in the domain of automated theorem proving.
>
> 2. **RL significantly enhances the sample efficiency of proof completion**
>
>     Given the steep difficulty levels of the benchmark problems, another meaningful metric for assessing model capability is the number of samples required to achieve a certain accuracy threshold. Referring back to Figure 3 in this paper, a key insight is highlighting the remarkable sample efficiency of the RL model. To achieve $50\\%$ accuracy on the miniF2F benchmark, the RL model requires approximately $32$ samples, demonstrating a 4-fold improvement compared to the $128$ samples needed by the SFT model. This conclusion also applies to large-scale experiments incorporating MCTS-based approaches. As demonstrated in Table 3, the SFT model requires over $16 \times 6400$ samples to achieve $60\\%$ accuracy on the miniF2F-test benchmark, whereas the RL model achieves the same level of accuracy with just approximately $4 \times 6400$ samples, attaining a 4-fold improvement in sample efficiency.
>
>
> **Question #1: GRPO vs. PPO**
>
> To evaluate the effectiveness of PPO compared to GRPO, we conduct PPO experiments using hyperparameters aligned with those used for GRPO in the paper. The results indicate that PPO does not outperform the GRPO baseline.
>
> | Pass@128 | GRPO (CoT) | PPO (CoT) | GRPO (non-CoT) | PPO (non-CoT) |
> | --- | --- | --- | --- | --- |
> | MiniF2F-test | $51.6\\%\pm 0.5\\%$ | $51.2\\%\pm0.4\\%$ | $50.5\\%\pm 0.6\\%$ | $50.4\\%\pm0.7\\%$ |
> | ProofNet-test | $18.2\\%\pm 0.5\\%$ | $17.2\\%\pm 0.7\\%$ | $17.5\\%\pm 0.5\\%$ | $16.7\\%\pm0.6\\%$ |
>
> GRPO is a simplified implementation of PPO, widely adopted in numerous LLM projects for its computational efficiency. The exclusion of the value network is motivated by the observation that learning a value function is often not simpler than addressing the core policy learning task, particularly for language tasks with trajectory-level sparse rewards. In policy gradient algorithms, the major functionality of the value function is to reduce the variance of gradient estimation. The choice of any arbitrary value function, as long as it depends solely on the state, preserves the unbiased estimation of the policy gradient. Since a learned value function often deviates significantly from the true value function, as noted in prior studies (Ilyas et al., 2020), using group-level average return as a Monte Carlo estimation serves as a satisfactory approximation.
>
> *References*
>
> [2] Lample et al. (2022). Hypertree proof search for neural theorem proving.
>
> [3] Xin et al. (2024). DeepSeek-Prover: Advancing theorem proving in LLMs through large-scale synthetic data.
>
> [4] Wu et al. (2024). Lean-Github: compiling Github Lean repositories for a versatile Lean prover.
>
> [5] Ilyas et al. (2020). A closer look at deep policy gradients.

---

> ### Author Response · Authors · 2024-11-24
> **Response to Reviewer ePCd (4/5)**
>
> **Question #2: Why not generate one tactic at a time, as is standard in multi-step RL?**
>
> From a theoretical perspective, generating a complete proof in one sequence is equivalent to generating it step-by-step, as both approaches ultimately compose the same final proof. Our approach adopts the whole-proof generation paradigm based on several practical engineering considerations.
>
> 1. One major purpose of our truncation mechanism is reducing the communications between LLM generation (GPU jobs) and Lean verification (CPU jobs). Leveraging the underlying infrastructure for LLM inference, generating a complete sequence in a single request is significantly faster than breaking it down into multiple steps. In a multi-step MDP setup, where Lean verification is required at each proof step, each step would involve transferring the generated tokens from the GPUs to CPUs, decoding them into UTF-8 proof codes, transmitting the data to the Lean server, waiting for the verification response, and then re-encoding the proof assistant's feedback into LLM tokens before transferring them back to the GPUs for further processing. This process introduces significant overhead to the proof generation pipeline.
> 2. Breaking down the proof generation into step-by-step processes does not offer additional extrinsic supervision for reinforcement learning. The only extrinsic reward is a binary signal indicating whether the entire proof is complete or not. Unlike classical control tasks in traditional deep RL, there is no intermediate reward signal to guide the generation of individual proof steps. Given the sparse-reward nature of the task and the equivalence in downstream data processing, we develop our algorithmic pipeline upon the whole-proof generation model.
>
> **Question #3: Am I correct that given an equivalent sample budget the difference between MCTS and 1-shot generation methods is simply the way candidate solutions are generated? Does the reduced number of new tokens generated result in meaningful compute savings for the MCTS algorithm?**
>
> Yes. Given an equivalent sample budget, the tree search approach and 1-shot generation differ only in their prompt sampling strategies, while the total number of model invocations remains the same. For the second question, the answer is also yes: the use of MCTS can effectively reduce the number of output tokens. We compare the average output tokens generated by our RL model across four settings: (1) with and without RMaxTS; (2) with and without chain-of-thought. The results show that nearly half of the output tokens can be saved through tree search, as the generation leverages existing nodes with partial code.
>
> | avg. tokens / sample | CoT | non-CoT |
> | --- | --- | --- |
> | single-pass generation | 495.7 | 109.6 |
> | RMaxTS | 244.6 | 76.1 |

---

> ### Author Response · Authors · 2024-11-24
> **Response to Reviewer ePCd (5/5)**
>
> **Question #4: Figure 4 needs to define what the shaded regions are, especially since the regions are heavily overlapping, leading me to question the statistical significance of the benefits of RMaxTS over the ablations.**
>
> The shaded regions in Figure 4 represent the confidence interval, corresponding to one standard deviation around the mean accuracy. The experimental results in Figure 4 contain two parts. The left panel presents the experimental outcomes for a sample budget of fewer than $6400$ samples, where the performance difference between the RMaxTS algorithm and its ablation version is approximately one standard deviation. As the sample size increases, the right panel highlights a more pronounced performance gap. With a larger sample budget, i.e., $4\times 6400$, the difference between RMaxTS and the ablation version widens, reaching up to two standard deviations. This demonstrates a clear improvement in sample efficiency for RMaxTS, as it scales more effectively than the baseline methods.
>
> **Question #5: Why do some accuracy numbers have +- intervals and others don't?**
>
> In this paper, we have prioritized making the statistical results more rigorous by providing confidence intervals alongside our key findings. For all experiments with manageable sample sizes, the average accuracy $\mu$ and the corresponding standard deviation $\sigma$ are computed by running 16 independent trials (using 16 random seeds) to replicate the sampling and evaluation process. Specifically, for the miniF2F dataset, experiments with a sample budget below $4\times 6400$ are based on 16 independent runs, while those with a budget of exactly $4\times 6400$ are estimated from 4 independent runs. For the ProofNet dataset, the threshold is set at $3200$. The results of ProofNet with a sample budget below $3200$ are derived from 16 independent runs, while those with a sample budget of exactly $3200$ are estimated from 4 independent runs. Due to computational constraints, we do not perform multiple trials for experiments with the largest sample budgets. In the paper revision, we include a detailed description in Appendix B.1 to clarify the computation of confidence intervals.
>
> Additionally, we would like to highlight that the results from smaller sample sizes are sufficient to demonstrate significant improvements over baseline models. For instance, on the miniF2F benchmark, our RL model achieves an accuracy of $54.9\\%\pm0.7\\%$ with a sample size of $3200$, which is comparable to the strongest baseline, InternLM2-StepProver, that requires a substantially larger sample budget of $64 \times 3200$ to reach an accuracy of $54.5\\%$. This underscores the superior sample efficiency of our model.

---

> > ### Comment · Reviewer_ePCd · 2024-11-26
> > **Response to Rebuttal**
> >
> > Thanks for your detailed explanations! On reflection, I think my judgement of the algorithmic novelty was too harsh, given the domain-specific nature of this method. As someone not working on this or a related topic, I'll defer to others regarding whether the significance of the problem domain justifies that degree of specificity, since I can't reasonably evaluate it.
> >
> > That said, I do still have some reservations about the significance of the results. I acknowledge that these benchmarks are very difficult, but it seems to me that the bulk of the performance improvement over prior work is present in the DS-Prover-V1.5-SFT baseline, and while MCTS and RL both help a little their impact is much smaller. DS-Prover-V1.5-SFT adds chain of thought reasoning and tactic state information to the prompt, but (If I understand correctly) this all comes down to prompt engineering rather than novel training or sampling strategies.
> >
> > Simple extensions can be worth disseminating if they have significant impact, but I'm not sure how to weigh that for this domain since it is unfamiliar to me (how big of a deal is this for automated theorem proving? This could be further emphasized in the motivation possibly?), so I will set it aside and defer to other reviewers on that point. Regardless, the bulk of this paper is concerned with the two bigger extensions, RL fine tuning and MCTS, which improve performance only a relatively small amount. If these extensions only provide a modest uplift, it seems odd for the paper to focus on them rather than on the benefits of chain-of-thought and tactic state information.
> >
> > As such, while I'm increasing my score I'm reluctant to recommend acceptance without a perspective on this significance from the other reviewers (especially since the other reviews also noted the small size of the performance improvements). Hopefully we can have some useful discussion during the reviewer discussion period on this point.

---

> > > ### Author Response · Authors · 2024-11-27
> > > **Thanks for your constructive suggestions**
> > >
> > > Thanks for increasing the score and leading the discussion. We would like to provide further explanation to better interpret the experimental results.
> > >
> > > One notable feature worth highlighting is that **the SFT dataset comprises synthetic proofs generated by RL+MCTS.** Specifically, our SFT dataset is curated using an expert-iteration process, as adopted by many prior work on formal theorem proving, such as GPT-f (Polu & Sutskever, 2020) and InternLM2-StepProver (Wu et al., 2024). This iterative approach alternates between model training and proof data collection, continually enriching the SFT dataset with proofs validated through Lean verification, which is motivated by the fact that most open-source datasets provide only formal problem statements without accompanying Lean proofs. This procedure resembles the training paradigm of AlphaZero, though it is not implemented in a fully online manner. **The strong performance of our SFT model is essentially the result of imitating the MCTS policy.**  This paper exclusively relies on open-source datasets for expert iteration, without leveraging additional Lean problem statements (details in Appendix A.2). The significant performance improvement achieved over prior work employing a similar expert-iteration training paradigm underscores the effectiveness of RL and MCTS.
> > >
> > > We hope our response addresses the reviewer's concerns, and we would gladly engage in further discussions or answer any follow-up questions.
> > >
> > > *References*
> > >
> > > [1] Polu & Sutskever (2020). Generative language modeling for automated theorem proving.
> > >
> > > [2] Wu et al. (2024). Lean-Github: compiling Github Lean repositories for a versatile Lean prover.

---

### Meta-Review · Area_Chair_QLsb · 2024-12-23

**Metareview:**

This paper uses proof assistant feedback to improve the capabilities of LLMs to construct formal proofs in Lean. It casts the problem in the RL paradigm using Lean verification outcomes as the reward signal. The generation of diverse proofs is encouraged by employing a variant of MCTS designed to increase exploration of Lean tactics. The experiments show that these technique achieve competitive results on the miniF2F and ProofNet benchmarks.

**Additional Comments On Reviewer Discussion:**

The authors provided detailed responses to the concerns raised by the reviewers. While most reviewers concur that the bar for acceptance has been reached, reviewer ePCd provided a dissenting opinion with the key point being that the work seems rather incremental and insufficiently impactful. I thank the reviewer for engaging with the authors and providing suggestions for improving the paper. Having carefully reviewed the discussion and the paper, I feel that the authors have provided sufficient evidence to justify their claims, and that this method of integrating proof assistant feedback to enhance the theorem-proving capabilities of LLMs is sufficiently innovative as to warrant acceptance. In particular, a specific percentage improvement over baselines is not a criteria for acceptance and we should endeavor to judge the paper on the merits of the proposed method as a whole. As it stands, especially accounting for the scores provided by the other reviewers, the paper makes a significant enough contribution to meet the threshold for acceptance.

---

### Decision · Program_Chairs · 2025-01-22

Accept (Poster)